# Single cell transcriptional and chromatin accessibility profiling redefine cellular heterogeneity in the adult human kidney

Yoshiharu Muto [1,7], Parker C. Wilson [2,7], Nicolas Ledru [1], Haojia Wu[1], Henrik Dimke [3,4], Sushrut S. Waikar [5] & Benjamin D. Humphreys [1,6 ✉]

The integration of single cell transcriptome and chromatin accessibility datasets enables a deeper understanding of cell heterogeneity. We performed single nucleus ATAC (snATAC-seq) and RNA (snRNA-seq) sequencing to generate paired, cell-type-specific chromatin accessibility and transcriptional profiles of the adult human kidney. We demonstrate that snATAC-seq is comparable to snRNA-seq in the assignment of cell identity and can further refine our understanding of functional heterogeneity in the nephron. The majority of differentially accessible chromatin regions are localized to promoters and a significant proportion are closely associated with differentially expressed genes. Cell-type-specific enrichment of transcription factor binding motifs implicates the activation of NF-κB that promotes *VCAM1* expression and drives transition between a subpopulation of proximal tubule epithelial cells. Our multi-omics approach improves the ability to detect unique cell states within the kidney and redefines cellular heterogeneity in the proximal tubule and thick ascending limb.

[1] Division of Nephrology, Department of Medicine, Washington University in St. Louis, St. Louis, MO, USA. [2] Department of Pathology and Immunology, Washington University in St. Louis, St. Louis, MO, USA. [3] Department of Cardiovascular and Renal Research, Institute of Molecular Medicine, University of Southern Denmark, Odense, Denmark. [4] Department of Nephrology, Odense University Hospital, Odense, Denmark. [5] Section of Nephrology, Department of Medicine, Boston University School of Medicine and Boston Medical Center, Boston, MA, USA. [6] Department of Developmental Biology, Washington University in St. Louis, St. Louis, MO, USA. [7] These authors contributed equally: Yoshiharu Muto, Parker C. Wilson. ✉email: humphreysbd@wustl.edu

The kidney is composed of diverse cell types with distinct subpopulations and single-cell sequencing can dissect cellular heterogeneity at high resolution[1]. For example, a small subset of cells within the proximal tubule and Bowman's capsule express vimentin, CD24, and CD133[2,3]. These cells have a distinct morphology and expression profile and undergo expansion after acute kidney injury[3]. Furthermore, they have reported potential for tubular and podocyte regeneration[4–7] and have been implicated in the development of renal cell carcinoma[8]. However, the sparsity of these cells has hampered further characterization. Another example of cellular heterogeneity is seen in the thick ascending limb. Studies in mouse and human suggest that there are structural and functional differences between medullary and cortical thick ascending limb, however, the signaling pathways that drive these differences are not well defined[9].

Single-cell or nucleus RNA sequencing (scRNA-seq or snRNA-seq) has fostered a greater understanding of the genes and pathways that define cell identity in the kidney[1]. Multiple scRNA-seq atlases of mature human[10–13] and mouse kidney[14,15] have established how transcription contributes to cell-type specificity. Recent methods have expanded this approach to single-cell profiling of chromatin accessibility[16–18]. Single nucleus assay for transposase-accessible chromatin using sequencing (snATAC-seq) is an extension of bulk ATAC-seq[19] that employs hyperactive Tn5 transposase to measure chromatin accessibility in thousands of individual cells[17]. Chromatin accessibility is a dynamic process that drives nephron development and nephron progenitors have distinct chromatin accessibility profiles that change as they differentiate[18]. The role of chromatin accessibility in the promotion or inhibition of kidney repair and regeneration has important implications for designing therapies for acute and chronic kidney disease and may help to improve directed differentiation of kidney organoids[20]. Joint profiling by scRNA-seq and snATAC-seq in the adult mouse kidney has provided a framework for understanding how chromatin accessibility regulates transcription[16]; however, the single-cell epigenomic landscape of the human kidney has not been described.

Integration and analysis of multimodal single-cell datasets is an emerging field with enormous potential for accelerating our understanding of kidney disease and development[21]. Bioinformatics tools can extract unique information from snATAC-seq datasets that is otherwise unavailable by scRNA-seq. Prediction of cell-type-specific cis-regulatory DNA interactions[22] and transcription factor activity[23] are two methods that complement the transcriptional information obtained by scRNA-seq. Long-range chromatin-chromatin interactions play an important role in transcriptional regulation and are influenced by transcription factor binding[24]. Chromatin accessibility profiling will help to identify distant regulatory regions that influence transcription via long-range interactions.

We have performed snATAC-seq and snRNA-seq to examine how chromatin accessibility can refine our understanding of cell state and function in the mature human kidney. We generated an interactive multimodal atlas encompassing both transcriptomic and epigenomic data (http://humphreyslab.com/SingleCell/). Data tracks for the cis-coaccessibility networks and cell-specific differentially accessible chromatin are available for download and viewing with the UCSC genome browser (https://genome.ucsc.edu/s/parkerwilson/control_celltype_cr). Combined snRNA-seq and snATAC-seq analysis improved our ability to detect unique cell states within the proximal tubule and thick ascending limb and redefines cellular heterogeneity that may contribute to kidney regeneration and cell-specific cation permeability.

## Results

**Single-cell transcriptional and chromatin accessibility profiling in the adult human kidney.** snRNA-seq and snATAC-seq were performed on five healthy adult kidney samples (Fig. 1a). Selected patients ranged in age from 50 to 62 years and included men ($n = 3$) and women ($n = 2$). All patients had preserved kidney function (mean sCr = 1.07 mg/dl, eGFR = 64.4 ± 4.7 ml/min/1.73 m²). Histologic review showed no significant glomerulosclerosis or interstitial fibrosis and tubular atrophy (Supplementary Table 1). We processed these samples at different timepoints. Batch correction was performed with the R package "Harmony" for both snATAC-seq and snRNA-seq datasets. We used Seurat to determine the cellular composition of our snRNA-seq samples, annotate cells based on their transcriptional profiles, and inform our snATAC-seq analysis (Fig. 1a). snRNA-seq identified all major cell types within the kidney cortex (Fig. 1b, Supplementary Fig. 1a) based on the expression of lineage-specific markers (Fig. 1c, Supplementary Fig. 1b). We detected proximal tubule (PT), parietal epithelial cells (PEC), thick ascending limb (TAL), distal tubule (DCT1, DCT2), connecting tubule (CNT), collecting duct (PC, ICA, ICB), endothelial cells (ENDO), glomerular cell types (MES, PODO), fibroblasts (FIB), and a small population of leukocytes (LEUK) (Supplementary Table 2, Supplementary Data 1). Notably, there was a subpopulation of proximal tubule that had increased expression of *VCAM1* (PT_VCAM1). This subpopulation also expressed *HAVCR1* (kidney injury molecule-1), which is a gene upregulated in the proximal tubule after acute injury and a predictor of long-term renal outcomes[25].

**Integration of single nucleus RNA and ATAC datasets for prediction and validation of ATAC cell-type assignments.** snATAC-seq captures the chromatin accessibility profile of individual cells[17]. Relatively less is known about cell-type-specific chromatin accessibility profiles; so we leveraged our annotated snRNA-seq dataset to predict snATAC-seq cell types with Seurat using label transfer[21]. Label transfer was performed by creating a gene-activity matrix from the snATAC-seq data, which is a measure of chromatin accessibility within the gene body and promoter of protein-coding genes. Transfer anchors were identified between the "reference" snRNA-seq dataset and "query" gene activity matrix followed by the assignment of predicted cell types. The distribution of snATAC-seq prediction scores showed that the vast majority of cells had a high prediction score and were confidently assigned to a single-cell type (Supplementary Fig. 2). The snATAC-seq dataset was filtered using a 97% confidence threshold for cell-type assignment to remove heterotypic doublets. Comparison between snATAC-seq cell-type predictions obtained by label transfer (Fig. 1d) and curated annotations of unsupervised clusters (Fig. 1e, f, Supplementary Fig. 1c, d and Supplementary Table 3) indicates that all major cell types were present in both datasets and that snATAC-seq is comparable to snRNA-seq in the detection and assignment of cell identities (Supplementary Fig. 3). We performed downstream analyses with gene-activity-based cell-type assignments, which were obtained by unsupervised clustering of the snATAC-seq dataset. Interestingly, snATAC-seq was able to detect two subpopulations within the proximal tubule cluster, which likely represent the proximal convoluted tubule (Fig. 1e, PCT) and the proximal straight tubule (Fig. 1e, PST). PCT showed greater chromatin accessibility in *SLC5A2*, which encodes sodium glucose cotransporter 2 (SGLT2); whereas PST showed greater accessibility in *SLC5A1* (Fig. 1f, Supplementary Fig. 4) which encodes the sodium glucose cotransporter 1 (SGLT1). SGLT2 reabsorbs glucose in the S1 and S2 segments of the proximal tubule and SGLT1 is located in S3[26].

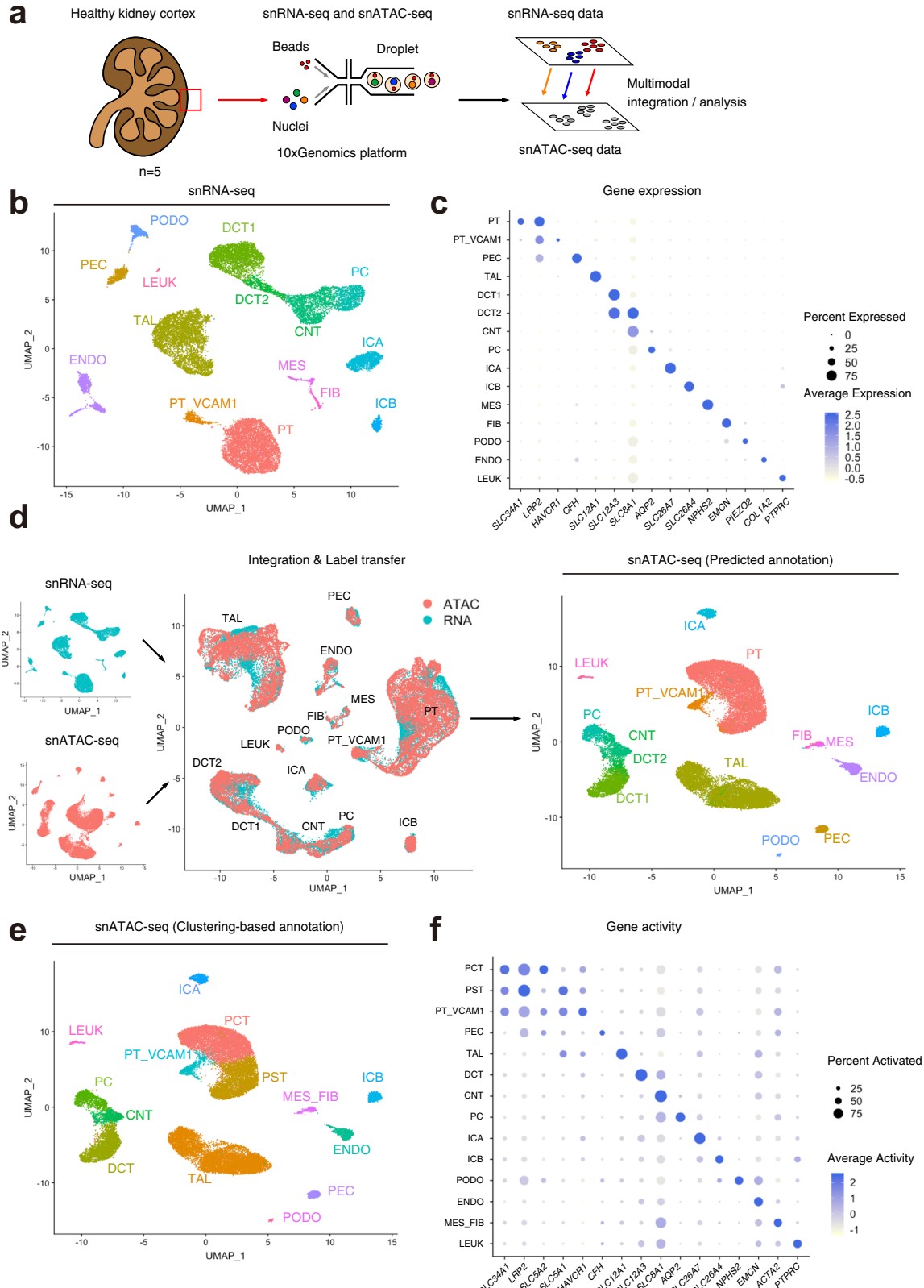

The delineation between S1/S2 and S3 was less clear in the snRNA-seq dataset (Supplementary Fig. 4), which suggests that snATAC-seq provides complementary information that may refine cell-type assignment; particularly for genes transcribed at low levels or genes that are not detected by snRNA-seq. To further illustrate this point, we down-sampled the snATAC-seq dataset so that it contained the same number of cells as the snRNA-seq dataset. The down-sampled snATAC-seq dataset retained the ability to distinguish between PCT and PST segments even after we further reduced the number of snATAC-seq cells to half the number of snRNA cells (Supplementary Fig. 5). The chromatin accessibility profile of S1/S2 may be of clinical interest

**Fig. 1 Single-cell transcriptional and chromatin accessibility profiling on the human adult kidneys. a** Graphical abstract of experimental methodology. $n = 5$ human adult kidneys were analyzed with snRNA-seq and snATAC-seq. **b** UMAP plots of snRNA-seq dataset. PT, proximal tubule; PT_VCAM1, subpopulation of proximal tubule with *VCAM1* expression; PEC, parietal epithelial cells; TAL, thick ascending limb; DCT, distal convoluted tubule; CNT, connecting tubule; PC, principle cells, ICA, Type A intercalated cells; ICB, Type B intercalated cells; PODO, podocyte; ENDO, endothelial cells; MES, mesangial cells, FIB, fibroblasts; LEUK, leukocytes. **c** Dot plot of snRNA-seq dataset showing gene expression patterns of cluster-enriched markers. The diameter of the dot corresponds to the proportion of cells expressing the indicated gene and the density of the dot corresponds to average expression relative to all cell types. **d** Multi-omics integration strategy for processing the snATAC-seq dataset. Following integration and label transfer, the snATAC-seq dataset was filtered using a 97% prediction score threshold for cell-type assignment. **e** UMAP plot of snATAC-seq dataset with gene activities-based cell-type assignments. PCT, proximal convoluted tubule; PST, proximal straight tubule. **f** Dot plot of snATAC-seq dataset showing gene-activity patterns of cell-type markers. The diameter of the dot corresponds to the proportion of cells with detected activity of indicated gene and the density of the dot corresponds to average gene activity relative to all cell types.

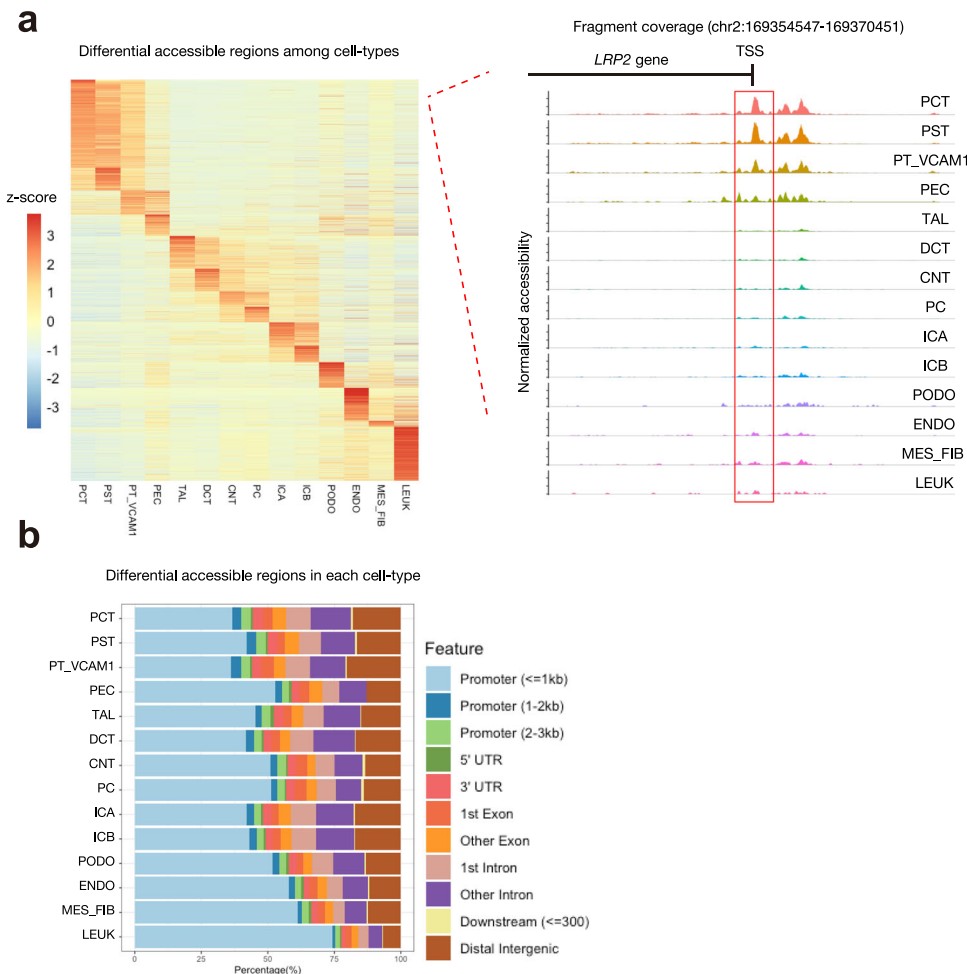

**Fig. 2 Distribution of cell-type-specific chromatin accessible regions. a** Heatmap of average number of Tn5 cut sites within a differentially accessible region (DAR) for each cell type (left). The color scale represents a z-score of the number of Tn5 sites within each DAR scaled by row. Fragment coverage (frequency of Tn5 insertion) around the DAR (DAR ± 50 Kb) on the *LRP2* gene promoter is shown (right). **b** Bar plot of annotated DAR locations for each cell type.

in determining the factors that drive glucose reabsorption, which is the therapeutic target of SGLT2 inhibitors[26]. Together, our multimodal snATAC-seq and snRNA-seq analysis improved our ability to dissect cellular heterogeneity.

**Chromatin accessibility defines cell type**. We detected 214,890 accessible chromatin regions among 27,034 cells in the snATAC-seq library. We compared these regions to a previously published dataset of DNase I-hypersensitive sites (DHS) in bulk glomeruli and tubulointerstitium[27]. DNase hypersensitivity is an alternative

measure of chromatin accessibility and approximately 50% of all regions identified by our pipeline were overlapping with a DHS in the glomerulus or tubulointerstitium. The proportion of overlapping regions increased to ~85% when our dataset was filtered for regions contained in at least 10% of nuclei (Supplementary Fig. 6). These data suggest that snATAC-seq is a robust method for the detection of accessible chromatin in the adult kidney.

We used the R package Signac[21] to investigate differences in chromatin accessibility between cell types. Cell types can be distinguished based on whether differentially accessible chromatin regions (DAR) are "open" or "closed" (Fig. 2a, Supplementary

Data 2). We employed a log-fold-change threshold of 0.25 and used Bonferroni-adjusted *p*-values to assess significance (padj < 0.05) of DAR. Approximately 20% (mean proportion = 0.203 ± 0.04) of DAR were closely associated with differentially expressed genes in their respective cell types (Supplementary Table 4). For example, *LRP2* is a lineage-specific gene expressed in the proximal tubule and a coverage plot in this region shows an increase in number and amplitude of ATAC peaks within its promoter and gene body (Fig. 2a). In fact, the majority of DAR were located in a promoter region within 3 kb of the nearest transcriptional start site (Fig. 2b, Supplementary Fig. 7. The second most common location was intronic and the distribution of DAR was relatively conserved across cell types (Fig. 2c). A minority of cell-type-specific differentially expressed genes were closely associated with a DAR (mean proportion = 0.358 ± 0.07), which raises the question of assigning function to DAR that are not located near differentially expressed genes. Regulatory regions can associate via long-range interactions and a DAR does not necessarily regulate the closest gene. Bioinformatics tools can infer regulatory chromatin interactions and may be useful for assigning function to DAR[22]. Long-range interactions mediate the association between enhancers and promoters via chromatin looping and are regulated in part by transcription factors[24].

**Chromatin accessibility is associated with cell-type-specific transcription factor activity and chromatin interaction networks**. Transcription factors are key determinants of cell fate that drive cellular differentiation in kidney aging and development[28]. Transcription factor "activity" can be predicted for individual cell types based on the presence of binding motifs within DAR. We used chromVAR[23] to infer transcription-factor-associated chromatin accessibility in our snATAC-seq dataset. We observed that individual cell types can be defined by transcription factor "activity" (Fig. 3a, Supplementary Data 3), suggesting that cell-type-specific transcription factors likely regulate chromatin accessibility. *HNF4A* encodes a key transcription factor that drives proximal tubule differentiation[29]. chromVAR detected an enrichment of HNF4A binding motifs within DAR in the proximal tubule (Fig. 3b, motif activity) that was supported by increased chromatin accessibility in *HNF4A* (Fig. 3b, gene activity) and increased *HNF4A* transcription in the snRNA-seq dataset (Fig. 3b, gene expression). We validated HNF4A binding by chromatin immunoprecipitation followed by quantitative PCR (ChIP-qPCR) to the predicted HNF4A binding sites within DAR for selected target gene loci (*SLC34A1, SLC5A2, HNF4A*) in primary renal proximal tubule epithelial cells (RPTEC, Supplementary Fig. 8a). HNF4A expression was detectable in RPTEC, however, at a lower level than kidney cortex, suggesting robust interactions of HNF4A and these loci in this cell type (Supplementary Fig. 8b).

A similar pattern was seen for *TFAP2B*, which regulates development in the distal nephron[30]. There was increased *TFAP2B* transcription factor "activity" in the thick ascending limb and distal convoluted tubule (Fig. 3b, motif activity), in addition to increased chromatin accessibility in *TFAP2B* (Fig. 3b, gene activity) and increased *TFAP2B* transcription (Fig. 3b, gene expression). The AP-2 family of transcription factors consists of five proteins in mice and humans encoded by *TFAP2A, TFAP2B, TFAP2C, TFAP2D,* and *TFAP2E*[31] with a well-established role in kidney development[30,32]. To further explore the role of transcription factors in determining distal nephron fate, we performed pseudotemporal ordering of the distal convoluted tubule (DCT), connecting tubule (CNT), and principal cells (PC), which form a distinct cluster of transcriptionally related cell types in both the snRNA and snATAC datasets (Fig. 1). We identified

pseudotime-dependent chromatin regions that distinguish between DCT and PC while progressing from proximal to distal along the distal nephron (Supplementary Fig. 9a, b) and performed a transcription factor motif enrichment within these regions with Signac. We identified 24 transcription factor motifs that were significantly enriched (FDR < 0.05) with a fold enrichment greater than two. TFAP2B was among those transcription factors which also included a couple of candidate transcription factors that potentially regulate distal nephron fate (*ZBTB33, CREB3, E2F1*). Subsequently, we performed pseudo-temporal ordering of the distal nephron cells in snRNA dataset to identify pseudotime-dependent gene modules that change their expression pattern from proximal to distal (Supplementary Fig. 9c, d). We intersected the genes in the gene module with the highest activity in PC and CNT with enriched transcription factor motifs to identify transcription factors that change their motif activity as well as their expression (Supplementary Fig. 9e, f). As a result, we identified additional members of the AP-2 family (*TFAP2A, TFAP2C*) and several candidate transcription factors (*EGR1, ZSCAN4, STAT1, STAT3, KLF9, NR2F2, IRF1*) that may also play a role in determining distal nephron fate.

We used the R package Cicero[22] to predict cis-regulatory chromatin interactions for individual cell types. Cis-coaccessibility networks (CCAN) are families of chromatin regions that co-vary their accessibility and can be used to predict chromatin interactions. Within *HNF4A*, we observed a robust CCAN in the proximal convoluted tubule with multiple connections (red or blue arcs) between differentially accessible regions (Fig. 3c, red boxes) in the promoter, gene body, and distal regions (Fig. 3c). Overall, we observed poor correlation between Cicero gene activity and gene expression (Pearson $r^2 = 0.12$), but good agreement with public databases of known chromatin-chromatin interactions. We compared our predicted interactions with the GeneHancer database[33] to determine which connections had been previously-reported in the literature. GeneHancer is a collection of human enhancers and their inferred target genes created using four methods: promoter capture Hi-C, enhancer-targeted transcription factors, expression quantitative trait loci, and tissue co-expression correlation between genes and enhancer RNA. The subset of GeneHancer interactions with "double elite" status is the most stringent set of interactions in the database[33] and the majority of predicted Cicero interactions within 50 kb of a cell-type-specific DAR were overlapping with "double elite" GeneHancer interactions (Fig. 3c, blue arcs). The proportion of Cicero connections present in the "double elite" GeneHancer database was dependent on the Cicero coaccessibility score, which is a measure of increased confidence of the predicted interaction (Supplementary Fig. 10). The Cicero connections with a lower coaccessibility score were less likely to be in the GeneHancer "double elite" database compared to Cicero connections with a higher coaccessibility score ($p < 2.2 \times 10^{-16}$, chi-squared). Within the proximal convoluted tubule, the majority of Cicero connections were either within a promoter region or between a promoter and another location (Fig. 3d) and this distribution was similar in other cell types (Supplementary Fig. 11). Data tracks for the cis-coaccessibility networks and cell-specific differentially accessible chromatin are available for download and viewing with the UCSC genome browser (Supplementary Fig. 12).

On a global level, transcription factor activity had modest correlation with transcription factor expression (Pearson $r^2 = 0.36$, *p*-value = $4.2 \times 10^{-12}$, Fig. 3e), however, transcription factors can act as either activators or repressors depending on cell type and context. We categorized transcription factors into three groups: (1) those that showed significant positive correlation between motif activity and gene expression ($n = 38$), 2) those that showed negative correlation ($n = 11$) and 3) those that showed no

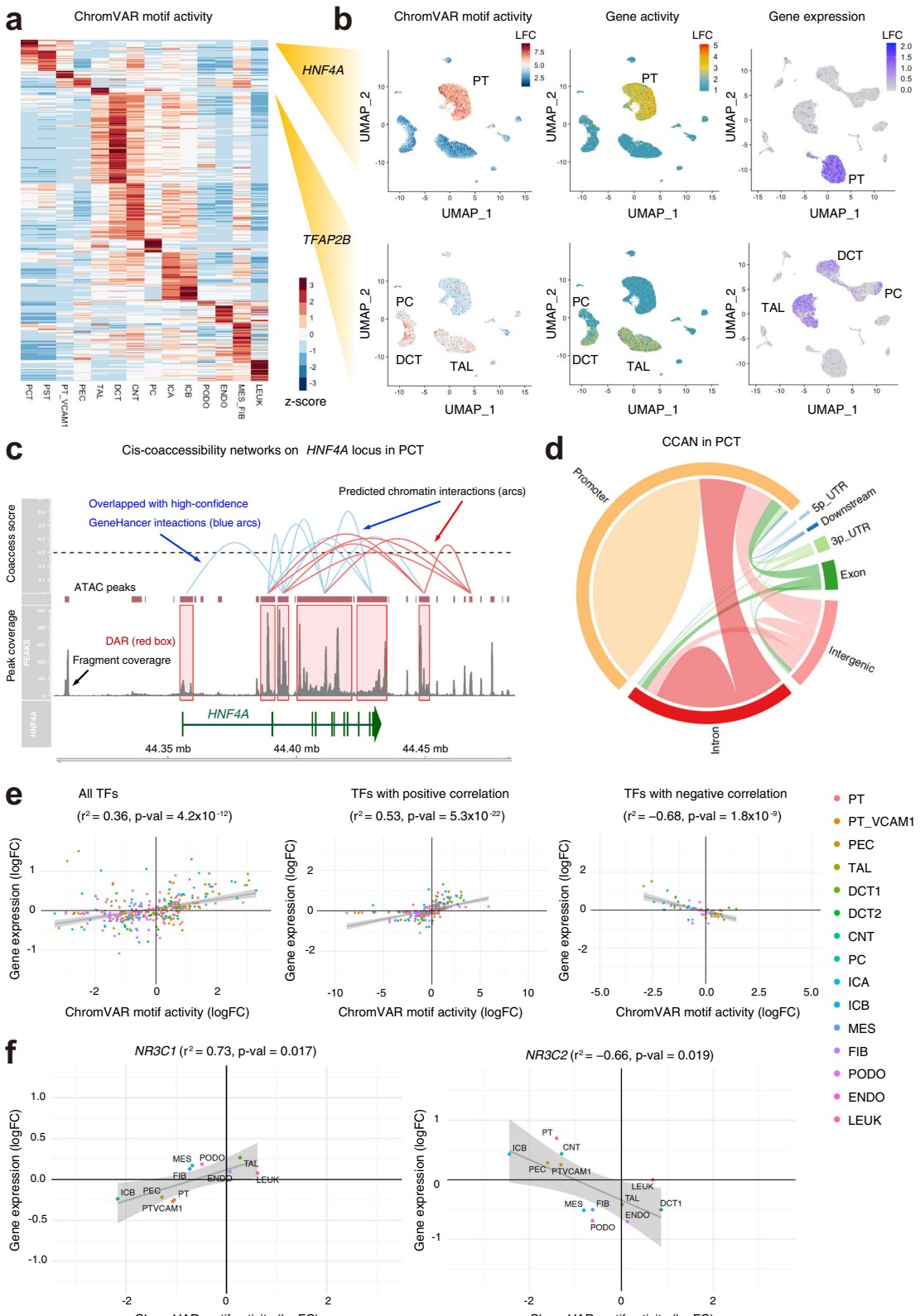

correlation ($n = 403$). We hypothesize that transcription factors with positive correlation between motif activity and transcription factor expression may be acting as transcriptional activators within DAR and those with negative correlation are acting as transcriptional repressors. Surprisingly, glucocorticoid receptor (*NR3C1*) showed a positive correlation between motif activity and

expression whereas mineralocorticoid receptor (*NR3C2*) showed a negative correlation (Fig. 3f). NR3C1 and NR3C2 are closely-related steroid hormones with nearly-identical binding motifs that have different biologic functions in the nephron[34] and this opposing relationship between motif activity and expression may regulate cell-specificity.

**Fig. 3 Cell-type-specific transcription factor activity and chromatin interaction networks. a** Heatmap of average chromVAR motif activity for each cell type. The color scale represents a z-score scaled by row. **b** UMAP plot displaying chromVAR motif activity (left), gene activity (middle) and gene expression (right) of *HNF4A* or *TFAP2B*. The color scale for each plot represents a normalized log-fold-change (LFC) for the respective assay. **c** Cis-coaccessibility networks (CCAN, red or blue arcs) near the *HNF4A* locus in the proximal convoluted tubule (PCT) with multiple connections between DAR (red boxes). DAR overlapping with high-confidence GeneHancer interactions are shown as blue arcs. Fragment coverage (frequency of Tn5 insertion) and called ATAC peaks are shown in the lower half. *HNF4A* gene track is shown along the bottom of the image. **d** Circos plot displaying CCAN in the PCT. **e** Cell-specific mean chromVAR motif activity from the JASPAR database was plotted against cell-specific average log-fold-change expression for the corresponding transcription factor for all cell types and transcription factors (left), transcription factors with significant positive correlation (middle) and transcription factors with significant negative correlation (right). **f** Mean chromVAR activity was plotted against average log-fold-change for glucocorticoid receptor (*NR3C1*, left) and mineralocorticoid receptor (*NR3C2*, right). Significant correlation was assessed with Pearson's product moment correlation coefficient using the cor.test function in R. snATAC-seq cell types were assigned using the label-transferred annotations from the snRNA-seq Seurat object. Cell types without significant chromVAR activity or transcription factor expression as determined by the Seurat FindMarkers function were not included in the plots.

**Allele-specific expression at the single-cell level.** Genetic variation is one of many factors that influences gene expression patterns. Allele-specific expression (ASE) refers to the relative contribution of maternal and paternal alleles that can be used to identify cis-regulatory variants that underlie phenotypic differences in a population[35]. We used GATK[36] to discover heterozygous germline single nucleotide variants (SNV) that overlap with coding transcripts (mRNA) and examined allelic-bias between donors using ASEP[35] after employing the WASP pipeline[37] to mitigate mapping bias. We first aggregated all cells into a "pseudobulk" dataset and analyzed 401 genes after applying our SNV filtering criteria. Among these 401 genes, we identified 84 with evidence of ASE after adjustment for multiple comparisons (Supplementary Data 4, Benjamini–Hochberg padj < 0.05). A subset of genes with ASE in the pseudo-bulk dataset are known to have important functions in the kidney, including *CLCNKB* and *SLC12A3* (Supplementary Fig. 13). We de-multiplexed our pseudo-bulk mRNA dataset to examine ASE limited to the proximal tubule (PT). Within the PT, we examined 77 genes and identified 17 with ASE. The majority of genes with ASE in the PT were also identified in the pseudo-bulk dataset ($n = 12/17$), however, there were a limited number of genes that were unique to the PT ($n = 5/17$). These genes were predominantly from the UDP-glycosyltransferase family (*UGT1A10, UGT1A8, UGT1A9*), which are involved in the elimination of exogenous chemicals and by-products of endogenous metabolism[38]. Subsequently, we expanded our analysis to include intronic reads (pre-mRNA), which increased the number of analyzed genes to 1430. Among these 1430 genes in the pre-mRNA analysis, we identified 432 with ASE that were enriched for GO biological processes, including glucuronidation and sodium ion transport. A total of 68 ($n = 68/84$, 80%) ASE genes in the pseudo-bulk pre-mRNA analysis were also identified in the mRNA analysis. We de-multiplexed our pseudo-bulk pre-mRNA dataset to examine ASE in the proximal tubule where we analyzed 221 genes and identified 62 with ASE. Among these 62 genes, 16 ($n = 16/62$, 25%) were only identified in the PT-specific pre-mRNA analysis, which suggests that allele-specific bias may be enriched in specific cell types.

**Multimodal analysis highlights cellular heterogeneity in the thick ascending limb.** The thick ascending limb in the cortex and medulla regulates sodium chloride balance, urinary concentration, and calcium and magnesium homeostasis. The majority of divalent cations are reabsorbed in the cortical segment and are regulated by the expression of claudins[39]. To determine if we could detect subpopulations of cells with variable claudin expression patterns, we performed unsupervised clustering on the thick ascending limb in our snRNA-seq dataset (Fig. 4a) to identify three groups of cells. There was a group of cells (SLC12A1+UMOD+) that expressed thick ascending limb

markers (*CLDN16, KCNJ10,* and *PTH1R*): TAL1 and a second group that expressed another set of TAL specific markers such as *CLDN10*: TAL2 (Fig. 4b). The third group of cells was identified as ascending thin limb (ATL) based on the expression of previously published markers[10]. We used immunohistochemistry to validate that PTH1R and KCNJ10 were expressed in a subset of UMOD+ SLC12A1+ cells (Fig. 4c).

We analyzed the thick ascending limb cluster in the snATAC-seq dataset and identified three groups of cells that echoed our findings in the snRNA-seq dataset (Fig. 4d, e). Our findings suggest that thick ascending limb subpopulations can be defined by either transcription or chromatin accessibility profiles. We used the Seurat FindMarkers function and chromVAR to identify increased transcription factor motif activity for HNF1B or ESRRB in either TAL subset relative to the remaining TAL cells (Bonferroni padj < 0.05) (Fig. 4f). Subsequently, we used the Seurat FindMarkers function to identify DAR that differentiate thick ascending limb cell populations and performed a transcription factor motif enrichment of these DAR using the Seurat FindMotifs function (Fig. 4g). Interestingly, HNF1B was the most enriched motif in open chromatin regions in TAL1 whereas ESRRB was enriched in open regions in TAL2. ESRRB is an orphan nuclear receptor with a critical role in early development and pluripotency[40] and HNF1B is a homeodomain-containing transcription factor that regulates nephrogenesis. Pathogenic germline *HNF1B* variants are a known cause of autosomal dominant tubulointerstitial kidney disease with hypomagnesemia and hypercalciuria[41]. Collectively, our multimodal analysis demonstrates heterogeneity within the thick ascending limb at a transcriptomic and chromatin accessibility level and highlights transcription factors that likely contribute to these differences.

**NF-κB regulates the molecular signature of a subpopulation of proximal tubule that expresses *VCAM1*.** We detected a subset of proximal tubule cells that had increased expression and chromatin accessibility of *VCAM1*, which we designated PT_VCAM1 (Fig. 1). Immunofluorescence studies demonstrated *VCAM1* expression in a scattered distribution amongst proximal tubule epithelium (Fig. 5a). Our single-cell studies estimate that PT_VCAM1 represents ~2% of total cells and 6% of proximal tubular epithelium. We also confirmed that VCAM1+ tubular cells were observed in 4.19 ± 1.58% of LTL+ PT cells, whereas no VCAM1+ cells were detected in UMOD+ TAL cells in the kidney cortex (Fig. 5b). Although previous studies have shown VCAM1 expression in the descending limb of loop of Henle (dTL), we only observed VCAM1 expressed in a subset of the dTL by costaining kidney sections with AQP1 (Fig. 5c). These data suggest that the majority of VCAM1+ tubular cells are in the proximal tubules within the cortex. Given that a small minority of dTL tubules expressed VCAM1 compared to VCAM1+ PT cells,

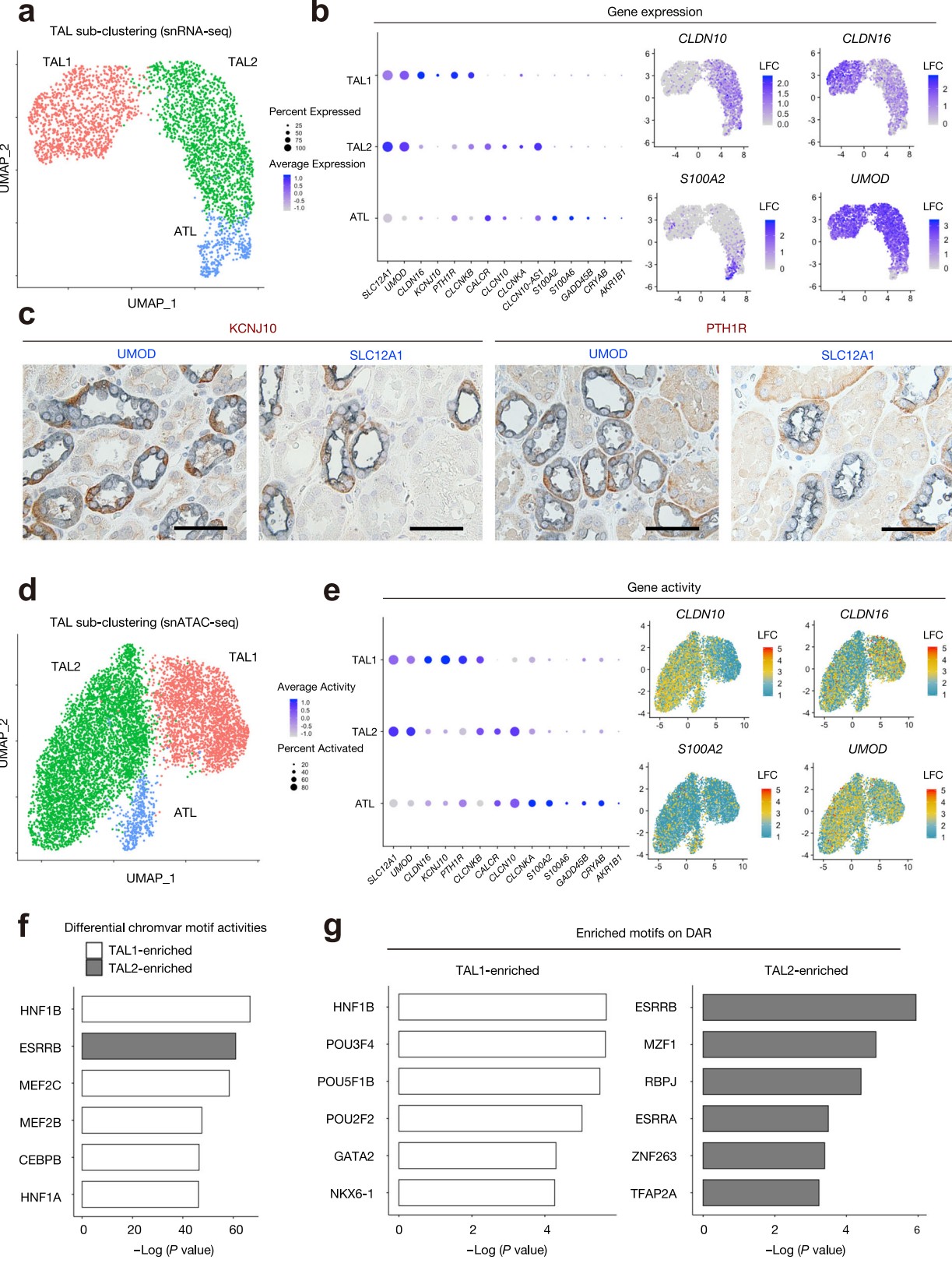

we concluded that VCAM1+ dTL cells are unlikely to be a significant portion of the cells in our dataset. Despite the fact that kidney samples originated from patients without kidney injury, the PT_VCAM1 population showed increased expression of kidney injury molecule-1 (KIM1, *HAVCR1*), which is a biomarker that is increased in acute kidney injury and chronic

kidney disease[42]. Interestingly, PT_VCAM1 also expressed *VIM* (vimentin), *CD24*, and *CD133* (Supplementary Fig. 14), which is consistent with a previously described population of cells with progenitor-like features in the proximal tubule[2–4]. We determined that a subset of VCAM1+ proximal tubular cells express CD24 or CD133 (Fig. 5d) in immunofluorescence analysis. We

**Fig. 4 Transcriptional and epigenetic heterogeneity in the thick ascending limb. a** Sub-clustering of TAL on the umap plot of snRNA-seq dataset to divide three subpopulations (TAL1, TAL2, and ATL). ATL, Ascending thin limb (of loop of Henle). **b** Dot plots showing gene expression patterns of the genes enriched in each of TAL subpopulations (left). The diameter of the dot corresponds to the proportion of cells expressing the indicated gene and the density of the dot corresponds to average expression relative to all cell types. Umap displaying gene expressions of *CLDN10*, *CLDN16*, *S100A2* or *UMOD* (right). **c** Representative immunohistochemical images of KCNJ10 or PTH1R (brown) and UMOD or SLC12A1 (blue) in the adult human kidneys. Scale bar indicates 50 μm. *n* = 3 samples were independently analyzed and similar results were obtained. **d** Sub-clustering of TAL on the umap plot of snATAC-seq dataset to divide three subpopulations (TAL1, TAL2, and ATL). **e** Dot plots showing gene activity patterns of the genes enriched in each of TAL subpopulations (right). The diameter of the dot corresponds to the proportion of cells with detected activity of indicated gene and the density of the dot corresponds to average gene activity relative to all cell types. Umap displaying gene activities of *CLDN10*, *CLDN16*, *S100A2*, or *UMOD* (left). **f** Differentially activated transcription factor motifs with chromVAR between TAL1 and TAL2. The top 6 motifs with the lowest *P* values are listed. **g** Motif enrichment analysis on the DARs between TAL1 and TAL2. Background was set as the genomic regions that are accessible to at least 2.5% of the TAL cells. The top 6 motifs with the lowest *P* values in each subpopulation are listed.

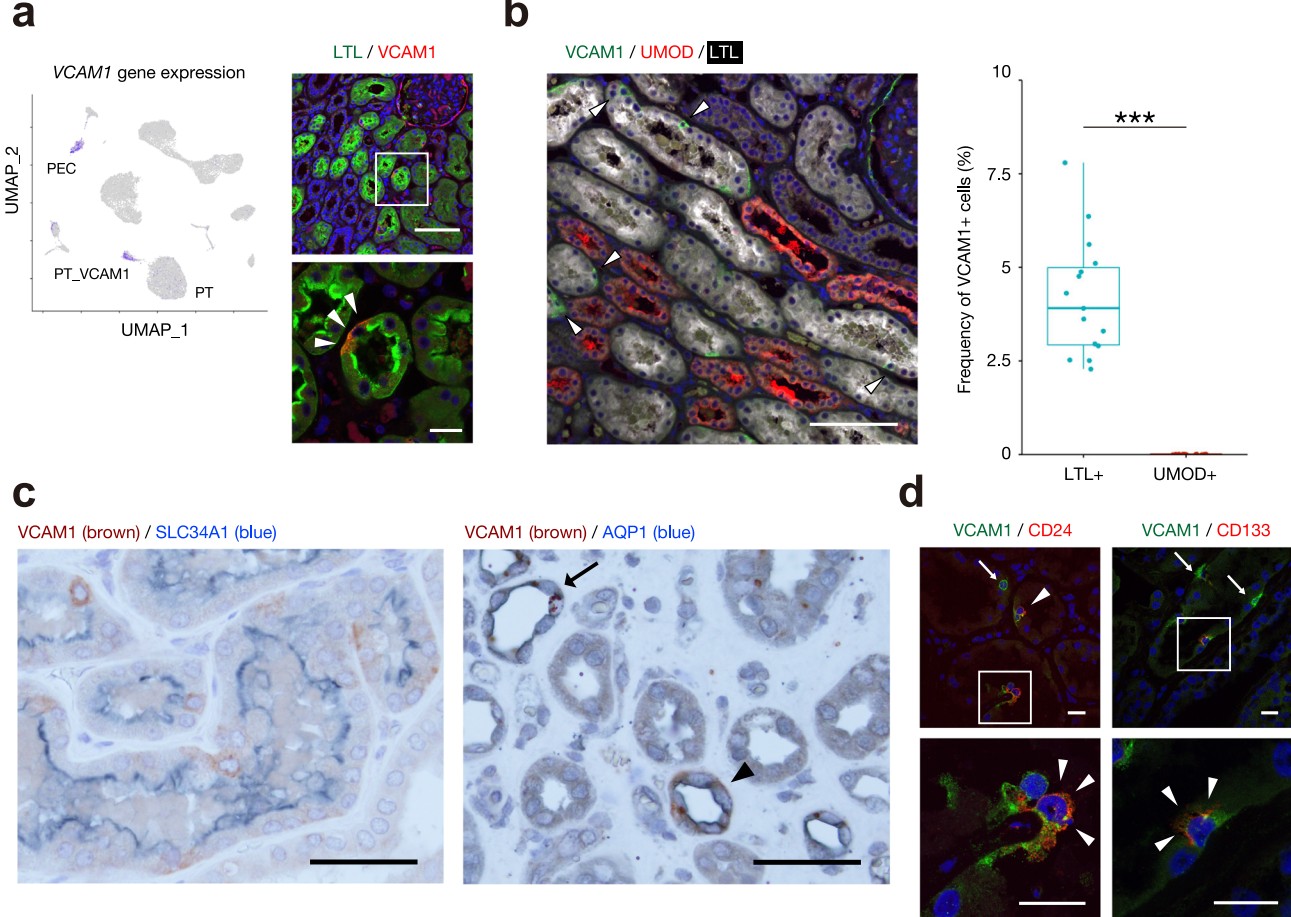

**Fig. 5 Identification of a subset of proximal tubular cells that express *VCAM1*. a** Umap plot displaying *VCAM1* gene expression in the snRNA-seq dataset (left), and representative immunohistochemical images of VCAM1 (red) or LTL (Lotus tetragonolobus lectin, green) in the adult kidney (*n* = 3 patients). Arrowheads indicate the VCAM1+ proximal tubular cells. VCAM1 was expressed in PEC and a subpopulation of LTL+ proximal tubular cells. Scale bar indicates 100 μm (upper right) or 20 μm (lower right). **b** Immunofluorescence staining for VCAM1 (green), UMOD (red) and LTL (white) in the adult human kidney sections (left, representative image) and quantitation of VCAM1-positive cells on the LTL-positive cells or UMOD-positive cells (right). The quantification was performed in five 200x images randomly taken from each patient (*n* = 3 patients). Arrowheads indicate VCAM1-positive cells in the LTL-positive PT. Scale bar indicates 100 μm. Box-and-whisker plots depict the median, quartiles and range. ***$P$ < 0.001 ($P$ = 5.47 × 10$^{-11}$, two-sided Student's *t* test). **c** Representative immunohistochemical images of SLC34A1 or AQP1 (blue) and VCAM1 (brown) in the adult human kidneys. An arrowhead marks VCAM1 expression in the DTL and an arrow marks DTL without VCAM1 expression. Scattered brown dots are seen with multiple different antibodies and considered non-specific staining. Scale bar indicates 50 μm. *n* = 3 samples were independently analyzed and similar results were obtained. **d** Representative immunostaining images of CD24 or CD133 (red) and VCAM1 (green) in the adult human kidney. Arrowheads indicate VCAM1 co-expression with CD24 or CD133 in PT and arrows mark VCAM1 expression without CD24 or CD133. Scale bar indicates 20 μm. *n* = 3 samples were independently analyzed and similar results were obtained.

also observed that CD24+ or CD133+ cells are rare in the VCAM1- proximal tubular cells. These findings indicate that the PT_VCAM1 cluster is heterogeneous and may represent an injured or regenerative subpopulation of cells.

We compared the transcriptional profile of PT_VCAM1 to the remaining proximal tubule to identify 464 differentially expressed genes with an absolute log-fold-change of at least 0.25 (FDR < 0.05, Supplementary Data 5). Gene-ontology enrichment analysis

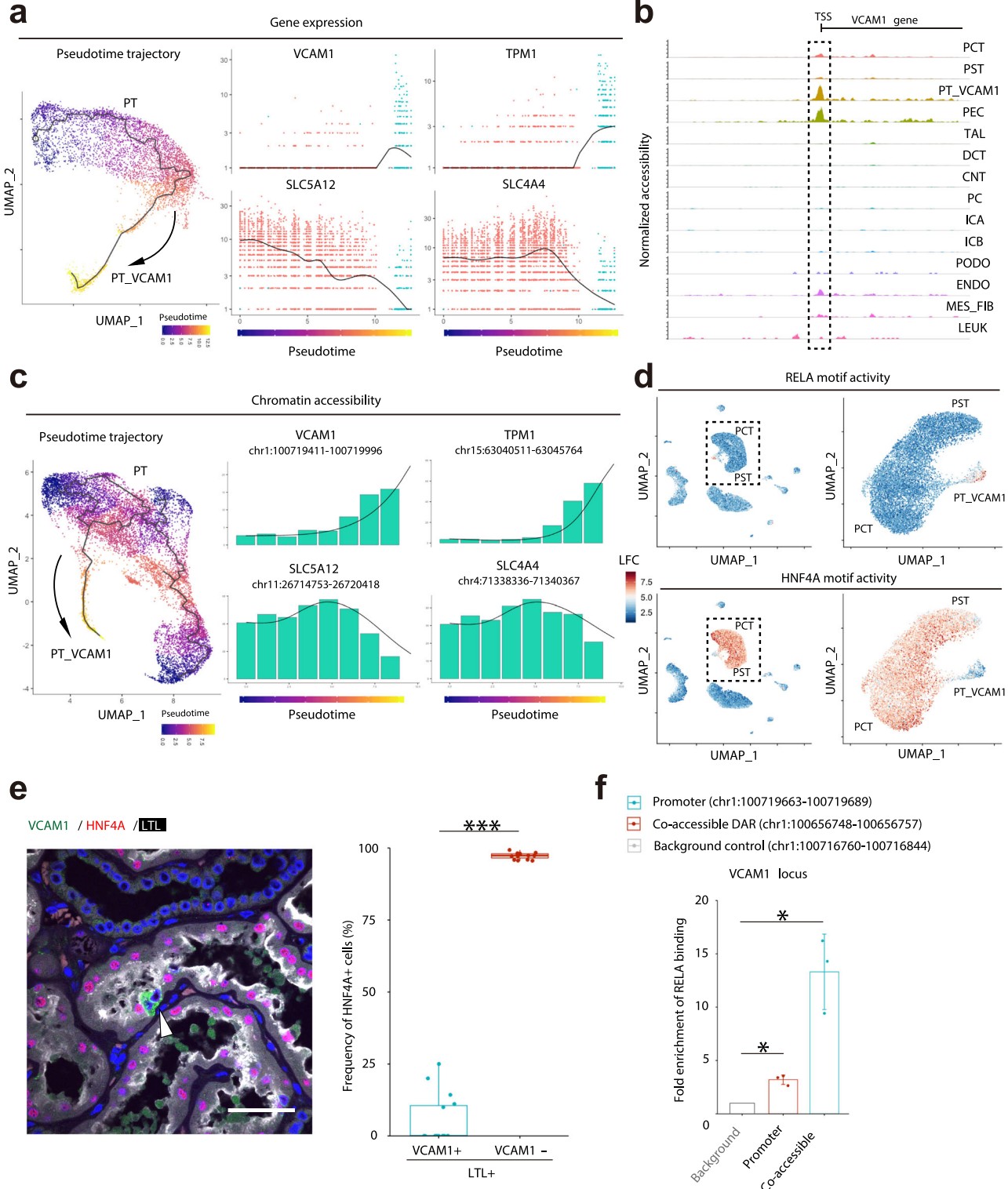

of the differentially expressed genes showed an enrichment for pathways involved in metabolism, cell migration, angiogenesis, proliferation, and apoptosis. In particular, there was enrichment for genes that control branching morphogenesis of epithelial tubes and the MAPK and Wnt signaling pathways (Supplementary Data 6). These results suggest that the signaling pathways in this subpopulation are distinct from the remaining proximal tubule.

We performed pseudotemporal ordering with Monocle[43] to determine which genes drive the transition from healthy proximal

tubule to the PT_VCAM1 state (Fig. 6a). We identified *VCAM1* and *TPM1* as genes that show increased expression in PT_VCAM1 cells and *SLC5A12* and *SLC4A4* as genes that show decreased expression (Fig. 6a). *VCAM1* is a key mediator of angiogenesis and *TPM1* encodes tropomyosin 1, which is an actin-binding protein involved in the cytoskeletal contraction. In contrast, *SLC5A12* and *SLC4A4* encode a lactate and bicarbonate transporter. *SLC5A12* and *SLC4A4* are abundantly-expressed in the PT where *VCAM1* and *TPM1* can be detected in a subset of

**Fig. 6 Characterization of a subset of proximal tubular cells using a multi-omics approach. a** Pseudotemporal trajectory from PT to PT_VCAM1 using snRNA-seq was generated with Monocle3 (left), and gene expression dynamics along a pseudotemporal trajectory from PT to PT_VCAM 1 are shown (right); VCAM1 (upper left), TPM1 (upper right), SLC5A12 (lower left) and SLC4A4 (lower right). **b** Fragment coverage (frequency of Tn5 insertion) around the representative DAR (DAR ± 5000 bp) in VCAM1 locus. **c** Pseudotemporal trajectory from PT to PT_VCAM1 using snATAC-seq was generated with Cicero (left). Chromatin accessibility dynamics along the pseudotemporal trajectory from PT to PT_VCAM1 are shown (right). chr1:100719411-100719996 (VCAM1 promoter, upper left); chr15:63040511-63045764 (TPM1 promoter, upper right), chr11:26714753-26720418 (SLC5A12 gene body, lower left) and chr4:71338336-71340367 (SLC4A4 gene body, lower right). **d** Feature plot of single-cell chromVAR motif activity of RELA and HNF4A in the entire dataset or PT/PT_VCAM1 subset. The color scale for each plot represents a normalized log-fold-change (LFC). **e** Immunofluorescence staining for VCAM1 (green), HNF4A (red) and LTL (white) in the adult human kidney sections (left, representative image) and quantitation of HNF4A-positive cells on the VCAM1-positive or negative subset of LTL-positive PT cells (right). The quantification was performed in five 200x images randomly taken from each patient ($n = 3$). Arrowheads indicate VCAM1-positive cells without HNF4A expression. Scale bar indicates 50 μm. Box-and-whisker plots depict the median, quartiles and range. \*\*\*$P < 0.001$ ($P = 1.21 \times 10^{-26}$, two-sided Student's $t$ test). **f** ChIP followed by quantitative PCR (ChIP-qPCR) analysis of RELA binding within the promoter or the open chromatin region that was predicted to interact with a VCAM1 promoter via a CCAN in the VCAM1 locus in RPTEC ($n = 3$ independent samples). The background control was set on the region without RELA motif at the upstream of VCAM1 promoter. See also Supplementary Fig. 17b (graphical method). Data are means ± s.d. \*$P < 0.05$ ($P = 0.0129$ and 0.0264, two-sided one sample $t$ test).

cells. We constructed a complementary pseudotemporal trajectory with Cicero[22] to examine changes in chromatin accessibility during the transition from PT to PT_VCAM1. Increased transcription of VCAM1 and TPM1 (Fig. 6a) was associated with increased chromatin accessibility within the VCAM1 gene body and promoter region (Fig. 6b, c). Similarly, decreased transcription of SLC5A12 and SLC4A4 (Fig. 6c) was associated with decreased chromatin accessibility (Fig. 6c, Supplementary Fig. 15). We identified transcription factors that likely regulate the transition between proximal tubule and PT_VCAM1 by assessing chromVAR transcription factor activities. Interestingly, the proximal tubule showed robust activity of HNF4A, which was decreased in the PT_VCAM1 cluster and coincided with increased activity of REL and RELA (Fig. 6d). We validated reduced HNF4A protein expression in PT_VCAM1 nuclei (Fig. 6e). NF-κB is a family of inducible transcription factors that share homology in the Rel domain that has been implicated in the inflammatory response in renal disease[44]. In particular, ischemia-reperfusion injury-induced acute kidney injury activates NF-κB and NF-κB inhibition improves renal function[45]. Consistent with this finding, gene set enrichment analysis of the differentially expressed genes in PT_VCAM1 compared to PT implicated NF-κB signaling (Supplementary Fig. 16). Interestingly, cultured RPTEC express VCAM1 (Supplementary Fig. 17a), likely reflecting injury/dedifferentiation as a consequence of in vitro culture. We asked whether cultured RPTEC could be used to validate the predicted role of RELA (NF-kB) in VCAM1 cis-regulatory interactions. We identified an open chromatin region ~60 kb from the VCAM1 gene body that contained a RELA motif predicted to interact with the VCAM1 promoter (via a cis-coaccessibility network). Indeed, this site was enriched for RELA binding by ChIP-qPCR (Fig. 6f, Supplementary Fig. 17b), providing experimental evidence for regulation of VCAM1 expression by RELA in this cell type.

**The proportion of PT_VCAM1 is elevated in acute kidney injury and chronic kidney disease**. We performed deconvolution of bulk RNA-seq obtained from mouse ischemia-reperfusion injury (IRI) experiments to determine if the proportion of PT_VCAM1 is related to acute kidney injury[46]. BisqueRNA estimates cell-type abundance from bulk RNA-seq using a scRNA-seq reference-based deconvolution[47]. The estimated proportion of PT_VCAM1 significantly increased 24 h post-IRI and persisted for at least 7 days; corresponding with a decrease in the proportion of normal proximal tubular cells (Fig. 7a). Interestingly, the estimated proportion of PT_VCAM1 in the no surgery control mouse kidneys increased in older mice (Fig. 7b)

and was accompanied by an increase in leukocytes. These results suggest a role for aging-related chronic inflammation and tubular injury in the appearance of PT_VCAM1. To further characterize the role of PT_VCAM1 in acute kidney injury, we used a snRNA-seq mouse IRI[48] dataset to predict the corresponding cell type for PT_VCAM1 in the injured mouse kidney. Label transfer of cell-type annotations from mouse IRI to human indicates that the majority of PT_VCAM1 are related to the failed-repair proximal tubule cell (FR-PTC) population that we recently identified in the mouse (Fig. 7c)[48].

We retrieved bulk RNA-seq datasets of healthy and injured human kidneys to estimate the proportion of PT_VCAM1[47]. We identified 72 non-tumor kidney samples in The Cancer Genome Atlas (TCGA) using the GDC data portal. The TCGA patients had a mean age of 62.5 years (s.d. = 11.9 years) and had undergone nephrectomy for renal cell carcinoma. Deconvolution of the non-tumor kidney samples with BisqueRNA[47] estimated that the proportion of PT_VCAM1 cells was 2.6% (Fig. 7d), which is consistent with our snRNA-seq and snATAC-seq estimates. Next, we analyzed bulk RNA-seq from kidney biopsies of patients with type 2 diabetes[49]. The patients with advanced diabetic nephropathy had a significantly higher proportion of PT_VCAM1 compared to control or early diabetic nephropathy patients (Fig. 7e), suggesting that PT_VCAM1 and tubular injury may be related to disease progression in diabetic nephropathy.

## Discussion
We performed snRNA-seq and snATAC-seq sequencing in parallel to describe the transcriptional and chromatin accessibility landscape of the adult human kidney. Our analysis demonstrates that snRNA-seq and snATAC-seq are comparable methods for determining cell identity and cell-type-specific chromatin accessibility provides additional information that further elucidates cellular heterogeneity. Multimodal single-cell profiling ("multi-omics") has greatly improved our ability to detect unique cell types and states while introducing a host of bioinformatics challenges and opportunities. In this study, we outline our integration approach to analyzing paired snRNA-seq and snATAC-seq datasets to highlight functional heterogeneity in the proximal tubule and thick ascending limb.

Studies in mouse and human suggest that there are structural and functional differences between thick ascending limb cells driven by regional expression patterns of claudins and other transport proteins[9]. Claudins are a family of tight junction proteins that confer segment-specific cation permeability and regulate the reabsorption of $Na^+$, $K^+$, $Cl^-$, $Mg^{2+}$ and $Ca^{2+}$. Claudin-10 (CLDN10) expression is enriched in the medullary

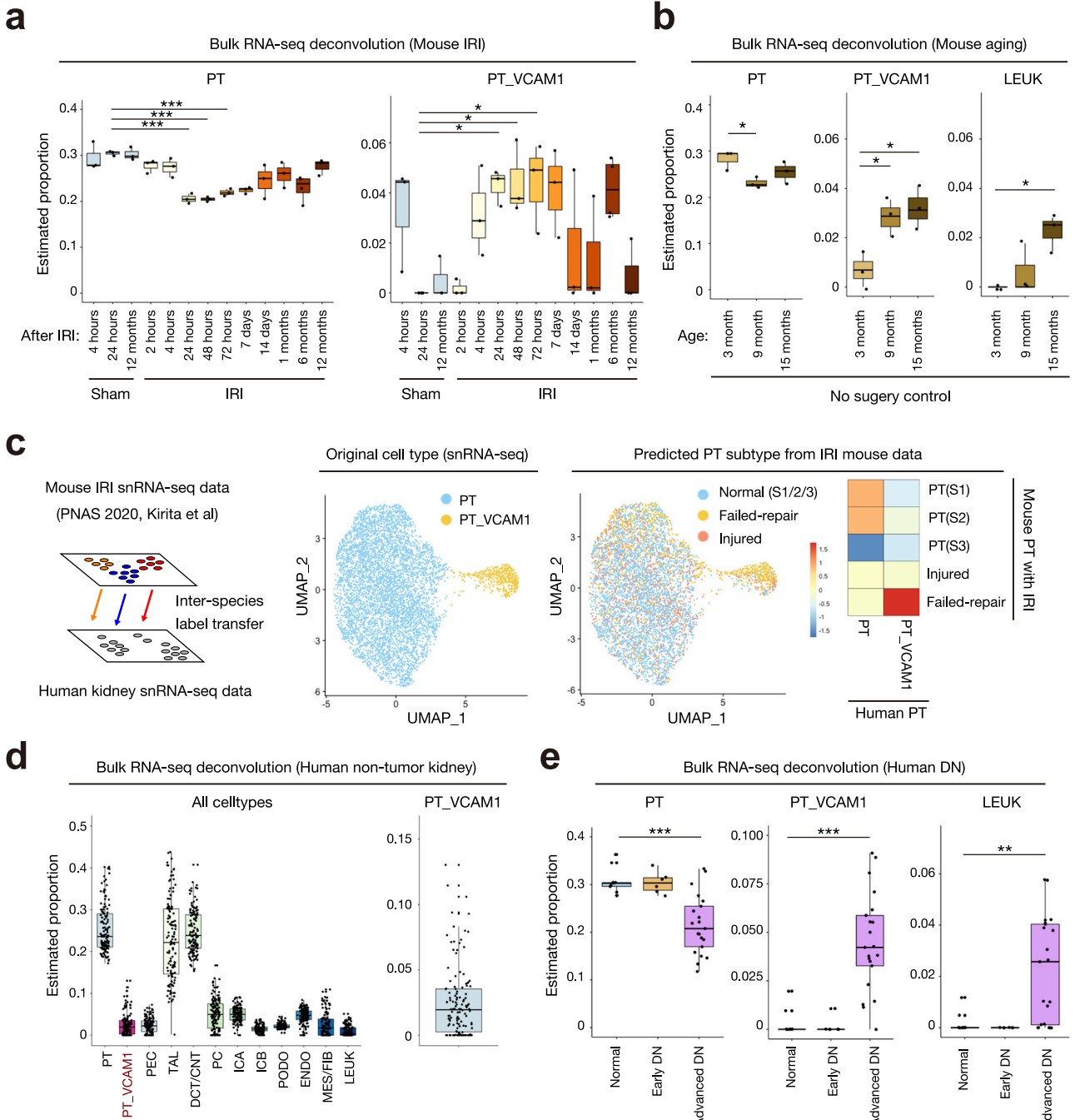

**Fig. 7 The estimated proportion of VCAM1+ proximal tubular cells increases in acute and chronic kidney disease. a, b** Deconvolution analysis of bulk RNA-seq mouse kidney IRI dataset (GSE98622) with BisqueRNA. Sham control and IRI (**a**), or no surgery control (**b**). **c** Inter-species data integration was performed between mouse IRI snRNA-seq (GSE139107) and human snRNA-seq with Seurat (left). PT and PT_VCAM1 from human snRNA-seq (middle) are label-transferred from mouse IRI snRNA-seq, and the frequencies of predicted cell types are shown on the heatmap (right). **d** Deconvolution analysis of bulk RNA-seq TCGA non-tumor kidney data (**e**) Deconvolution analysis of bulk RNA-seq human diabetic nephropathy (DN) data (GSE142025) with BisqueRNA. Box-and-whisker plots depict the median, quartiles and range. *$P < 0.05$; **$P < 0.01$; ***$P < 0.005$, one-way ANOVA with post hoc Dunnett's multiple comparisons test. All $P$ values are provided in the Data Source file.

thick ascending limb, whereas claudin-16 (CLDN16) is expressed predominantly in the cortical thick ascending limb. Mice lacking *Cldn16* develop hypercalciuria and hypomagnesemia, which is similar to the phenotype of patients with familial hypomagnesemia with hypercalciuria and nephrocalcinosis (FHHNC) that carry pathogenic variants in *CLDN16*[50]. In contrast, targeted deletion of *Cldn10* in the thick ascending limb results in impaired paracellular sodium permeability and hypermagnesemia[51]. These

data suggest that *Cldn10* and *Cldn16* differentially affect cation permeability in the thick ascending limb and are supported by the observation that *Cldn10* and *Cldn16* are expressed in a mosaic pattern in mice[52]. We observed two distinct subpopulations of *UMOD+* cells in the thick ascending limb (*CLDN10+CLDN16−* and *CLDN10−CLDN16+*) with enrichment of *KCNJ10* and *PTH1R* in *CLDN10-CLDN16+* cells. In line with these findings, we found a mosaic expression pattern of KCNJ10 and PTH1R in

the human TAL. In motif enrichment analysis, the *CLDN10-CLDN16+* population had increased transcription factor activity of HNF1B (Fig. 4e, f). Pathogenic germline variants in *HNF1B* are causative of autosomal dominant tubulointerstitial kidney disease with hypomagnesemia and hypercalciuria[41]. Validation of these findings are limited by the absence of TAL cell lines. Some rodent medullary TAL cell lines have been established previously[53,54], but they did not maintain characteristics of mature TAL cells. Additionally, murine Hnf1b regulates Cldn10b, Cldn19, and Cldn3 differently in one of these cell models in vitro than it does in vivo in mouse kidney[55]. Future studies using kidney organoids may help to investigate the roles of enriched transcription factors including HNF1B in TAL function.

The proximal tubule is the most abundant cell type in the kidney cortex and is divided into segments (S1, S2, S3) with unique functions driven by segment-specific expression of various transporters, including SGLT1 and SGLT2. SGLT2 is a therapeutic target in diabetic nephropathy and the genes and signaling pathways that regulate SGLT2 expression may be of clinical interest. snRNA-seq detected *SLC5A1* (SGLT1) and *SLC5A2* (SGLT2) in the proximal tubule, but could not clearly distinguish between the S1/S2 segments that express SGLT2 and the S3 segment that expresses SGLT1. In contrast, snATAC-seq was able to separate the S1/S2 and S3 segments based on chromatin accessibility within the gene body and promoter of *SLC5A1* and *SLC5A2*. These data suggest that snATAC-seq may help to further refine segment-specific cell types; particularly those that are defined by genes transcribed at low levels or genes that are not detected by snRNA-seq. Furthermore, snATAC-seq can predict the transcription factors that drive cell-type-specificity, which may improve our understanding of kidney development and directed differentiation of kidney organoids. We used this approach to implicate NF-κB signaling in a subpopulation of proximal tubule epithelial cells.

We used snRNA-seq and snATAC-seq to identify a subpopulation of proximal tubule (PT_VCAM1) that expressed *VCAM1, HAVCR1* (KIM-1), *VIM* (vimentin), *CD133*, and *CD24*. The PT_VCAM1 population was also identified in bulk RNA-seq datasets from non-tumor TCGA kidney and human diabetic nephropathy[49]. The proportion of PT_VCAM1 increased in response to acute and chronic kidney injury in both mouse and human. CD133+CD24+ progenitor-like cells have been previously described in the human kidney in a scattered distribution[2,3] and VCAM1 (CD106) expression is present in CD133+CD24+ renal progenitors localized to Bowman's capsule[4]. A separate population of CD133+CD24+VCAM1− cells are localized to the proximal tubule and both CD133+CD24+VCAM1+ and CD133+CD24+VCAM1− cells can engraft in SCID mice to repopulate the tubular epithelium following acute tubular injury[4]. *VCAM1, VIM, CD133*, and *CD24* expression was enriched in the PT_VCAM1 cluster in our snRNA-seq dataset (Supplementary Fig. 14), which differs from the previously described CD133+CD24+VCAM1− renal progenitor population localized to the proximal tubule. We used immunofluorescence studies to demonstrate that VCAM1+ cells are present in a scattered distribution within the proximal tubule of human kidneys (Fig. 5a). Comparison of our human data to a mouse snRNA-seq acute kidney injury dataset[48] suggests that PT_VCAM1 is closely related to a population of "failed-repair" PT, which has a proinflammatory gene expression signature (Fig. 7c)[48]. *HAVCR1* expression in PT_VCAM1 suggests that PT_VCAM1 likely represents a subpopulation of proximal tubular cells that is undergoing injury in situ, and expands in aging and chronic kidney disease (Fig. 7b, d). Pseudotemporal ordering (Fig. 6) indicated that PT_VCAM1 exists along a continuum with PT (Fig. 6), further supporting the hypothesis that they represent

an injured cell state. Motif enrichment analysis showed that PT_VCAM1 had increased RELA transcription factor activity and ChIP-qPCR suggested that RELA may regulate *VCAM1* expression in an in vitro model of proximal tubule cells (Fig. 6f). Our findings suggest that NF-κB plays a role in the maintenance of PT_VCAM1, which may be of clinical interest in designing therapies for acute kidney injury. However, whether proximal tubule repair involves proliferation of a progenitor population or dedifferentiation of mature epithelium still remains controversial[56] and our own previous results do not support the existence of a fixed intratubular progenitor population[57–59].

We employed allele-specific expression analysis to identify allelic bias in a pseudo-bulk dataset consisting of all cell types and after restricting our analysis to the proximal tubule. Overall, we estimate that the proportion of genes with allele-specific variation in the kidney ranges from 20.9 to 30.2% when we analyzed coding transcript and pre-mRNA variants, respectively. This estimate is significantly larger than the 4.6% reported by Fan et al., which relied on bulk RNA-seq[35]. Notably, sequencing depth significantly affects the ability to detect ASE and our samples were sequenced to a mean depth of 377 million reads; whereas the dataset described in Fan et al. was sequenced to a median depth of 35 million. Furthermore, our dataset was obtained from a nuclear dissociation, used 5′ sequencing chemistry and had far fewer samples. Future studies may benefit from exploring ASE in single-cell datasets to identify allelic bias restricted to individual cell types.

An advantage of snATAC-seq is the ability to measure covariance between accessible chromatin sites to predict cis-regulatory interactions[22]. This approach can link putative regulatory regions with their target genes and has been applied to human pancreatic islets[60], acute leukemia[61], and multiple mouse tissues, including hippocampus[62], mammary gland[63], T-cells[64], and kidney among others[16,17]. In particular, genome wide association study (GWAS) risk loci can be linked to their target genes, which would complement the progress made using chromosome conformation capture (Hi-C). We generated CCAN that had significant overlap with a published database[33]. The remaining interactions may represent the unique chromatin interaction landscape of the kidney. We have made all of our data publicly-available and invite readers to explore cell-type-specific differentially accessible chromatin regions by uploading custom tracks to the UCSC genome browser (Supplementary Fig. 12) or visiting our interactive website (Supplementary Fig. 18).

The small sample size of this study does not adequately capture the expected heterogeneity of the general population. Furthermore, our study focused on kidney cortex and did not include samples from the medulla. Future studies would benefit from studying diseased kidneys to determine how chromatin accessibility changes with disease progression. Also, improvements in peak calling algorithms for snATAC-seq data will help to narrow the differentially accessible chromatin regions and identify additional peaks in less common cell types. Another consideration for future studies includes the validation of snATAC-seq data by epigenetic modulation approaches such as dCas9-DNMT1 or dCas9-KRAB to alter the chromatin accessibility and verify corresponding changes in gene expression. Despite these limitations, our single-cell multimodal atlas of human kidney redefines cellular heterogeneity of the kidney driven by cell-type-specific transcription factors. Our data enhances the understanding of human kidney biology and provides a foundation for future studies.

## Methods

**Tissue procurement**. Non-tumor kidney cortex samples were obtained from patients undergoing partial or radical nephrectomy for renal mass at Brigham and Women's Hospital (Boston, MA) under an established Institutional Review Board protocol approved by the Mass General Brigham Human Research Committee. All

participants provided written informed consent in accordance with the Declaration of Helsinki. Samples were frozen or retained in OCT for future studies. Histologic sections were reviewed by a renal pathologist and laboratory data was abstracted from the medical record. For immunohistochemical staining of tissue for light microscopy, tissue was retrieved from nephrectomised kidneys due to renal carcinoma from parts of the kidney that not affected by tumor growth. Kidney tissue was immersion fixed in 10% formalin for 3 h and placed in phosphate-buffered saline (PBS) until embedded in paraffin. Informed written consent was obtained from each patient and approved by the Biomedical Research Ethics Committee of Southern Denmark in accordance with the Declaration of Helsinki.

**Nuclear dissociation and library preparation**. For snATAC-seq, nuclei were isolated with Nuclei EZ Lysis buffer (NUC-101; Sigma-Aldrich) supplemented with protease inhibitor (5892791001; Roche). Samples were cut into < 2 mm pieces, homogenized using a Dounce homogenizer (885302-0002; Kimble Chase) in 2 ml of ice-cold Nuclei EZ Lysis buffer, and incubated on ice for 5 min with an additional 2 ml of lysis buffer. The homogenate was filtered through a 40-μm cell strainer (43-50040-51; pluriSelect) and centrifuged at 500g for 5 min at 4 °C. The pellet was resuspended, washed with 4 ml of buffer, and incubated on ice for 5 min. Following centrifugation, the pellet was resuspended in Nuclei Buffer (10× Genomics, PN-2000153), filtered through a 5-μm cell strainer (43-50005-03, pluriSelect), and counted. For snRNA-seq preparation, the RNase inhibitors (Promega, N2615 and Life Technologies, AM2696) were added to the lysis buffer, and the pellet was ultimately resuspended in nuclei suspension buffer (1× PBS, 1% bovine serum albumin, 0.1% RNase inhibitor)[65]. 10X Chromium libraries were prepared according to manufacturer protocol.

**Single nucleus RNA sequencing bioinformatics workflow**. Five snRNA-seq libraries were obtained using 10X Genomics Chromium Single Cell 5′ v2 chemistry following nuclear dissociation[65]. Three snRNA-seq libraries (patients 1-3) were prepared for a prior study GSE131882[13]. A target of 10,000 nuclei were loaded onto each lane. Libraries were sequenced on an Illumina Novaseq instrument and counted with cellranger v3.1.0 using a custom pre-mRNA GTF built on GRCh38 to include intronic reads. The read configuration for libraries 1–3 was 2 × 100 bp and the configuration for libraries 4–5 was 2 × 150 bp paired-end. The cDNA for snRNA libraries was amplified for 17 cycles. Datasets were aggregated with cellranger v3.1.0 without depth normalization. A mean of 377,573,305 reads (s.d. = 76,365,483) were sequenced for each snRNA library corresponding to a mean of 70,886 reads per cell (s.d. = 8633, Supplementary Table 5). The mean sequencing saturation was 81.4 ± 2.4%. The mean fraction of reads with a valid barcode (fraction of reads in cells) was 88.2 ± 5.9% (Supplementary Table 6). Subsequently datasets were preprocessed with Seurat v3.0.2[21] to remove low-quality nuclei (Features > 500, Features < 4000, RNA count < 16000, %Mitochondrial genes < 0.8, %Ribosomal protein large or small subunits < 0.4) and DoubletFinder v2.0.2[66] to remove heterotypic doublets (assuming 5% of barcodes represent doublets). The filtered library was normalized with SCTransform, and corrected for batch effects with Harmony v1.0[67] using the "RunHarmony" function in Seurat. After filtering, there was a mean of 3997 ± 930 cells per snRNA-seq library and a mean of 1674 ± 913 genes detected per nucleus. Number of genes per cell, number of UMIs per cell and fraction of mitochondrial genes per cell for each patient were shown in Supplementary Fig. 19. Clustering was performed by constructing a KNN graph and applying the Louvain algorithm. Dimensional reduction was performed with UMAP and individual clusters were annotated based on expression of lineage-specific markers. The final snRNA-seq library contained 19,985 cells and represented all major cell types within the kidney cortex (Supplementary Table 1, Supplementary Fig. 20). Differential expression between cell types was assessed with the Seurat FindMarkers function for transcripts detected in at least 20% of cells using a log-fold-change threshold of 0.25. Bonferroni-adjusted p-values were used to determine significance at an FDR < 0.05.

**Single nucleus ATAC sequencing bioinformatics workflow**. Five snATAC-seq libraries were obtained using 10X Genomics Chromium Single Cell ATAC v1 chemistry following nuclear dissociation. A target of 10,000 nuclei were loaded onto each lane. Libraries were sequenced on an Illumina Novaseq instrument and counted with cellranger-atac v1.2 (10X Genomics) using GRCh38. The read configuration for libraries 1–3 was 2 × 50 bp paired-end and the configuration for libraries 4-5 was 2 × 150 bp paired-end. Sample index PCR was performed at 12 cycles. Libraries were aggregated with cellranger-atac without depth normalization. A mean of 318,097,692 reads were sequenced for each snATAC library (s.d. = 54,357,210) corresponding to a mean of 12,946 fragments per cell (s.d. = 2,960, Supplementary Table 5). The mean sequencing saturation for snATAC libraries was 37.3 ± 2.2% and the mean fraction of reads with a valid barcode was 97.3 ± 1.2% (Supplementary Table 7). Subsequently datasets were processed with Seurat v3.0.2 and its companion package Signac v0.2.1 (https://github.com/timoast/signac)[21]. Low-quality cells were removed from the aggregated snATAC-seq library (peak region fragments > 2500, peak region fragments < 25000, %reads in peaks > 15, blacklist ratio < 0.001, nucleosome signal < 4 & mitochondrial gene ratio < 0.25) before normalization with term-frequency inverse-document-frequency (TFIDF). A fraction of reads in peaks, number of reads in peaks per cell and ratio

reads in genomic blacklist region per cell for each patient were shown in Supplementary Fig. 19. Dimensional reduction was performed via singular value decomposition (SVD) of the TFIDF matrix and UMAP. A KNN graph was constructed to cluster cells with the Louvain algorithm. Batch effect was corrected with Harmony[67] using the "RunHarmony" function in Seurat. A gene activity matrix was constructed by counting ATAC peaks within the gene body and 2 kb upstream of the transcriptional start site using protein-coding genes annotated in the Ensembl database. The gene activity matrix was log-normalized prior to label transfer with the aggregated snRNA-seq Seurat object using canonical correlation analysis. The aggregated snATAC-seq object was filtered using a 97% confidence threshold for cell-type assignment following label transfer to remove heterotypic doublets. The filtered snATAC-seq object was reprocessed with TFIDF, SVD, and batch effect correction followed by clustering and annotation based on lineage-specific gene activity. After filtering, there was a mean of 5408 ± 1393 nuclei per snATAC-seq library with a mean of 7538 ± 2938 peaks detected per nucleus. The final snATAC-seq library contained a total of 214,890 unique peak regions among 27,034 nuclei and represented all major cell types within the kidney cortex (Supplementary Table 2, Supplementary Fig. 20). Differential chromatin accessibility between cell types was assessed with the Signac FindMarkers function for peaks detected in at least 20% of cells using a likelihood ratio test and a log-fold-change threshold of 0.25. Bonferroni-adjusted p-values were used to determine significance at an FDR < 0.05. Genomic regions containing snATAC-seq peaks were annotated with ChIPSeeker[68] (v1.24.0) and clusterProfiler[69] (v3.16.1) using the UCSC database[70] on hg38.

**Comparison to previously published database of DNase hypersensitive sites**. Glomerulus and tubulointerstitial DNase hypersensitive sites (DHS) were downloaded in bed format from Sieber et al.[27] Glomerulus and tubulointerstitial DHS master lists were composed by merging the tissue-specific bed files, converting to a GRanges object with the GenomicRanges package[71] (v1.40.0), and collapsing the intervals with the reduce function. cellranger-atac peaks were filtered by the proportion of nuclei containing the snATAC-seq peak and subsequently overlapped with DHS sites.

**Estimation of transcription factor activity from snATAC-seq data**. Transcription factor activity was estimated using the final snATAC-seq library and chromVAR v1.6.0[23]. The positional weight matrix was obtained from the JASPAR2018 database[72]. Cell-type-specific chromVAR activities were calculated using the RunChromVAR wrapper in Signac v0.2.1 and differential activity was computed with the FindMarkers function (FDR < 0.05). Motif enrichment analysis was also performed on the differentially accessible regions with the FindMotif function.

**Generation of cis-coaccessibility networks with Cicero**. Cis-coaccessibility networks were predicted using the final snATAC-seq library and Cicero v1.2[22]. The snATAC-seq library was partitioned into individual cell types and converted to cell dataset (CDS) objects using the make_atac_cds function. The CDS objects were individually processed using the detect_genes() and estimate_size_factors() functions with default parameters prior to dimensional reduction and conversion to a Cicero CDS object. Cell-type-specific Cicero connections were obtained using the run_cicero function with default parameters.

**Construction of pseudotemporal trajectories with Monocle or Cicero**. Monocle3[43] was used to convert the snRNA-seq dataset into a cell dataset object (CDS), preprocess, correct for batch effects[73], embed with dimensional reduction and perform pseudotemporal ordering. Cicero[22] was used to generate pseudotemporal trajectories for the snATAC-seq dataset. First, the CDS was constructed from the peak count matrix in the Seurat object of aggregated snATAC-seq data with "make_atac_cds" function with binarize = F. Next, the cds was preprocessed (num_dim = 50), aligned to remove batch effect[73] and reduced onto a lower dimensional space with the "reduce_dimension" function (reduction_method = 'UMAP', preprocess_method = "Aligned"). After filtering low-quality cells, the cells were clustered (cluster_cells) and visualized. Subsequently, the dataset was subsetted for the PCT, PST and PT_VCAM1 clusters or distal nephron clusters to perform cell ordering with the learn_graph function. We used the "order_cell" function and indicated three of the most distant cells from PT_VCAM1 in PCT and PST as "start points" of the trajectories in PT analysis or the most distant point in DCT in distal nephron analysis. The data were visualized with "plot_accessibility_in_pseudotime" or "plot_cells" functions.

**Comparison of Cicero coaccessibility connections to GeneHancer database**. Cell-type-specific differentially accessible chromatin regions (DAR) were identified with the Signac[21] FindMarkers function using a log-fold-change threshold of 0.25 for peaks present in at least 20% of cells (FDR < 0.05). Cell-type-specific DAR was extended 50 kb up- and downstream to create bed files to query the UCSC table browser[74] using the GeneHancer interactions tracks[33]. GeneHancer interactions were compared to cell-type-specific Cicero connections to determine the mean proportion of overlap with increasing Cicero coaccess threshold.

**Genotyping and variant annotation with GATK pipeline**. snATAC libraries were genotyped using GATK (v4.1.8.1) best practices for germline short variant discovery[36]. Fastq files were aligned with cellranger-atac (v1.2) and duplicates were marked with GATK MarkDuplicates. Base quality was recalibrated with Base-Recalibrator using the hg38 GATK resource bundle files: dbsnp138, known_indels, high-confidence SNPs from 1000 G, and Mills and 1000 G gold standard indels and subsequently applied with ApplyBQSR. Variants were called with HaplotypeCaller, genotyped with GenotypeGVCFs, scored with CNNScoreVariants, and filtered with FilterVariantTranches using a –snp-tranche of 99.95. snRNA libraries were aligned with cellranger (v3.1) and duplicates were marked with GATK MarkDuplicates. RNA reads were processed with SplitNCigarReads followed by BaseRecalibrator, ApplyBQSR, HaplotypeCaller, and GenotypeGVCFs as described in the snATAC pipeline. snRNA variants were hard-filtered with GATK VariantFiltration using FS > 30, QD < 2. snRNA and snATAC vcfs were annotated with GATK Funcotator and merged into a joint genotype file using bcftools (v1.7).

**Allele-specific expression analysis**. snRNA libraries were aligned with cellranger (v3.1) and processed with the WASP (v0.3.4) pipeline[37] using the joint genotype vcf composed of variants that overlap gene bodies. We used the find_intersecting_snps.py script to identify snRNA cellranger-aligned reads that overlap with SNV identified by GATK. We remapped overlapping reads with STAR (v.2.7.5)[75] and filtered the remapped reads with the WASP script prior to merging into a processed bam file. GATK ASEReadCounter was used to perform allele-specific counting at heterozygous positions that overlapped coding transcripts (mRNA) or pre-mRNA regions. Allele-specific counts that overlapped coding transcripts were filtered for positions that contained a total depth > 20, a read depth > 5 for both alleles, and a minor allelic fraction > 0.05. ASE was analyzed with ASEP (v0.1.0)[35] using default settings for unphased haplotypes. Benjamini-Hochberg adjusted p-values were used to assess significance (padj < 0.05).

**Gene-ontology enrichment analysis**. Differentially expressed genes in the PT_VCAM1 cluster (compared to PT) were identified with the FindMarkers function using a log-fold-change threshold of 0.25 for the genes expressed in at least 20% of cells (FDR < 0.05). Gene-ontology enrichment was performed with PANTHER (http://geneontology.org/).

**Gene set enrichment analysis**. Differential expressed genes in PT_VCAM1 cluster (compared to PT) were identified with the FindMarkers function using a log-fold-change threshold of 0.05 for peaks present in at least 5% of cells (FDR < 0.05). The pre-ranked gene list was analyzed with GSEA v4.0.3 (Broad Institute).

**Deconvolution of bulk RNA-seq data**. For the TCGA (The Cancer Genome Atlas) dataset, HTseq counts and metadata were downloaded from the GDC data portal (portal.gdc.cancer.gov) by selecting "kidney", "TCGA", "RNA-seq", and "solid tissue normal". Bulk RNA-seq counts were normalized with DESeq2[76] (v1.30.1) and count matrices were deconvoluted with BisqueRNA[47] (v1.0.4) using snRNA-seq annotations. For the mouse ischemia-reperfusion dataset from Liu et al.[46], a normalized count matrix was downloaded from GSE98622 and converted to human annotations using biomaRt and ensembl prior to deconvolution with BisqueRNA with default parameters. For the diabetic nephropathy dataset[49], fastq files were downloaded from GSE142025, transcript abundance was quantified with Salmon using GRCh38, count matrices were imported to DESeq2 with tximport (v1.16.1), and data was normalized prior to deconvolution with BisqueRNA.

**Inter-species snRNA-seq data comparison**. The snRNA-seq dataset for human adult kidneys was converted to mouse annotations using biomaRt and ensembl, and integrated with a mouse IRI snRNA-seq dataset (GSE139107)[48] using the FindTransferAnchors function in Seurat. Mouse cell-type annotations were transferred to the human dataset.

**Cell culture**. RPTEC (Lonza) was cultured with Renal Epithelium Cell Growth Medium 2 (PromoCell). Cells were maintained in a humidified 5% $CO_2$ atmosphere at 37 °C. Experiments were performed on early passages (passage 2-3).

**ChIP-qPCR**. Chromatin was prepared from cultured RPTECs using the Magna ChIP A/G Chromatin Immunoprecipitation Kit (Sigma-Aldrich, 17-10085). Briefly, cells were fixed with 1% fresh formaldehyde for 10 min at room temperature, then quenched with glycine. After cell lysis and subsequent nuclei lysis, chromatin was sonicated (Covaris ME220; 390 s with peak power 70 W, duty factor of 5%, and 1000 cycles per burst). Immunoprecipitation was performed on sheared chromatin with anti-RELA (Sigma-Aldrich, 17-10060) and anti-HNF4A (abcam, ab181604) antibodies overnight at 4 °C. Antibody-chromatin conjugates were captured on magnetic beads and washed. Following reversal of crosslinks with added Proteinase K, DNA purification was performed with the kit's provided Spin Columns. iTaq Univeral SYBR Green (BioRad, 1725125) was used to perform qPCRs using the primers listed on Supplementary Data 7. Technical replicates were averaged and Ct values for each target were normalized to 1% input signal. The % input value for each sample was then normalized to a background control locus in

*SLC34A1* for the HNF4A ChIP and *VCAM1* for the RELA ChIP. A two-tailed one-sample t test was performed to determine the statistical significance of the calculated fold enrichment of putative TF-binding loci relative to the negative control.

**Immunofluorescence studies**. Formalin-fixed paraffin-embedded tissue sections were deparaffinized and underwent antigen retrieval. Sections were blocked with 1% bovine serum albumin, permeabilized with 0.1% Triton-X100 in PBS and incubated overnight with primary antibodies for VCAM1 (EPR5047; abcam; ab134047; 1/200), biotinylated Lotus Tetragonolobus Lectin (Vector Laboratories; B-1325; 1/200), UMOD (Bio-Rad; 8595-0054; 1/200) or HNF4A (H-1; Santa Cruz Biotechnology; sc-374229; 1/200). Alternatively, fresh frozen sections were fixed in cold acetone for 5 min. Sections were blocked with 1% bovine serum albumin and incubated 1 h with primary antibodies for CD24 (SN-3; Santa Cruz Biotechnology; sc-19585; 1/20), CD133 (AC133; Miltenyi Biotec; 130-090-422; 1/10). These sections were subsequently stained with secondary antibodies; Alexa Fluor 488 donkey anti-rabbit antibody (Jackson ImmunoResearch; 711-545-152; 1/200); Cy3 donkey anti-sheep antibody (Jackson ImmunoResearch; 713-165-147; 1/200); Cy3 goat anti-mouse antibody (Jackson ImmunoResearch; 115-165-003; 1/200); Alexa Fluor 647 strepavidin (Jackson ImmunoResearch; 016-600-084; 1/200). Sections were stained with DAPI (4′,6′- diamidino-2-phenylindole) and mounted in Prolong Gold (Life Technologies). Images were obtained by confocal microscopy (Nikon C2 + Eclipse; Nikon, Melville, NY). The quantification was performed in five 200 × images randomly taken from each patient (n = 3).

**Immunohistochemical staining of tissue for light microscopy**. Paraffin-embedded sections were processed and stained as published in detail previously[77]. Following tissue rehydration and subsequent antigen retrieval by boiling in Tris-EGTA buffer (TEG, 10 mM Tris, 0.5 mM EGTA, pH = 9.0), 0.6% $H_2O_2$ and 50 mM $NH_4Cl$ in PBS were added to block endogenous peroxidases and free aldehyde groups. Sections were incubated overnight at 4 °C with primary antibody (0.1% Triton X-100 in PBS) for SLC12A1 (Sigma-Aldrich; HPA014967; 1/100), VCAM1 (EPR5047; abcam; ab134047; 1/200), Aquaporin 1 (abcam; ab15080; 1/100), SLC34A1 antibody (Clone 16; not commercially available; undiluted culture supernatant[78], PTH1R (R&D systems; AF5709; 1/100), UMOD (Biotrend; BT85-9500-54; 1/1000) or KCNJ10 (Alamone; APC-035; 1/1000). These sections were then washed and incubated with horseradish peroxidase (HRP) conjugated secondary antibodies; Rabbit Anti-Goat Immunoglobulins HRP conjugated (Dako, P0449; 1/200); Goat Anti-Mouse Immunoglobulins HRP conjugated (Dako, P0447; 1/200); Goat Anti-Rabbit Immunoglobulins HRP conjugated (Dako, P0448; 1/200). HRP activity was visualized using the DAB + Substrate Chromogen System (K3467, DakoCytomation). For double labeling of tissue, sections were reboiled in TEG followed by a new incubation round with primary antibodies and secondary HRP anibodies. The Vector SG chromagen substrate (Vector Laboratories, Burlingame, USA) was utilized for the detection of the second antigen. Sections were counterstained with hematoxylin and mounted using Aqua-Mount® (Thermo Scientific). Light microscopy was carried out using an Olympus BX51 microscope (Olympus, Denmark).

**Statistical analysis**. No statistical methods were used to predetermine sample size. Experiments were not randomized and investigators were not blinded to allocation during library preparation, experiments, or analysis. ChIP-qPCR data (Fig. 6f, Supplementary Fig. 8a) are presented as mean±s.d. and were compared between groups with two-tailed one-sample Student's t test. Quantitative data for immunofluorescence analysis are presented as box-and-whisker plots depicting the median, quartiles and range, and were compared between groups with two-tailed Student's t test. Estimated proportion by deconvolution of RNA-seq data (Fig. 7a, b, e) were analyzed with one-way ANOVA with post hoc Dunnett's multiple comparisons test. A P value of <0.05 was considered statistically significant.

**Reporting summary**. Further information on research design is available in the Nature Research Reporting Summary linked to this article.

## Data availability
All relevant data are available from the corresponding authors on reasonable request. Sequencing data are deposited in GEO under accession number GSE151302. Previously published snRNA-seq data for three adult kidneys are available in GEO (GSE131882). Data tracks for cis-coaccessibility networks and cell-specific differentially accessible chromatin are available for download and viewing with the UCSC genome browser at (https://genome.ucsc.edu/s/parkercwilson/control_celltype_cr)[74] (Supplementary Fig. 12). Gene expression, ATAC peaks, gene activities and motif activities for each cell type are also available via our interactive website; Kidney Interactive Transcriptomics (http://humphreyslab.com/SingleCell/) (Supplementary Fig. 18). Additional interactive visualization and annotation of the snRNA-seq and snATAC-seq datasets are available through the cellxgene portal located at https://cellxgene.cziscience.com/collections/9b02383a-9358-4f0f-9795-a891ec523bcc. Source data are provided with this paper. Public data repositories used for our analyses include Ensembl http://useast.ensembl.org/index.html, GeneHancer https://www.genecards.org/, and JASPAR http://jaspar.genereg.net/. Gene-ontology enrichment was performed with PANTHER (http://geneontology.

org/). For the TCGA (The Cancer Genome Atlas) dataset, counts and metadata were downloaded from the GDC data portal (portal.gdc.cancer.gov) by selecting "kidney", "TCGA", "RNA-seq", and "solid tissue normal". Source data are provided with this paper.

## Code availability

All analysis code is available on GitHub at https://github.com/p4rkerw/Muto_Wilson_NComm_2020 and has been deposited in Zenodo at https://doi.org/10.5281/zenodo.4555693.

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

## Acknowledgements

The authors thank Inger Nissen at the University of Southern Denmark for expert technical assistance and Dr. Niels Marcussen and Dr. Reza Zamini for collection of kidney material for immunohistochemical evaluation. These experiments were funded by seed network grant CZF2019-002430 from the Chan Zuckerberg Initiative (to B.D.H. and S.S.W.). Additional support was from the CKD Biomarkers Consortium Pilot and Feasibility Studies Program (NIDDK: U01 DK103225, to B.D.H.), the International Research Fund for Subsidy of Kyushu University School of Medicine Alumni, the Japan Society for the Promotion of Science (JSPS) Postdoctoral Fellowships for Research Abroad, a Postdoctoral Fellowship of the Uehara Memorial Foundation and a grant from the Nakatomi Foundation (all to Y.M.) and grant support from the Independent Research Fund Denmark (to H.D.).

## Author contributions

Y.M. and P.C.W. contributed equally to this study. Y.M., P.C.W., and B.D.H. planned the study. S.S.W. collected the kidney samples. Y.M. prepared the snRNA-seq and snATAC-seq libraries. Y.M., P.C.W., and B.D.H. analyzed and interpreted snRNA-seq and snATAC-seq data. Y.M. performed other experiments. N.L. performed and analyzed ChIP-qPCR. H.D. performed immunohistochemical analysis. H.W. generated the public database for data. Y.M., P.C.W., and B.D.H. wrote the manuscript. B.D.H. coordinated and oversaw the study. All authors discussed the results and commented on the manuscript.

## Competing interests

The authors declare no competing interests.
