## [Peer Review File · Nature Communications]

REVIEWER COMMENTS

Reviewer #1 (Remarks to the Author):

In this manuscript, Muto et al., utilized single nucleus RNA sequencing (snRNA-seq) and single nucleus ATAC sequencing (snATAC-seq) simultaneously to study the transcriptional and chromatin accessibility landscape of the adult human kidney. Joint profiling by scRNA-seq and snATAC-seq provided a framework for understanding how chromatin accessibility regulates transcription and this is the first study describing the single-cell epigenomic landscape of the human kidney. The authors applied this integration approach to highlight functional heterogeneity in the proximal tubule and thick ascending limb. They demonstrate that snATAC-seq provided additional information as compared to scRNA-seq and snATAC-seq.

The authors applied previously established R packages such as "chromVAR" to infer transcription-factor-associated chromatin accessibility, which means individual cell types are classified by transcription factor activity. The results are in agreement with previous finding that HNF4A binding motifs are enriched within differentially accessible chromatin regions (DAR) in the proximal tubule. Next, the authors applied the R package "Cicero" to predict cis-regulatory chromatin interactions for individual cell types. In order to study the cellular heterogeneity in the thick ascending limb, the authors performed unsupervised clustering analysis on the thick ascending limb in snRNA-seq dataset and identified 3 groups of cells: cortical thick ascending limb (CLDN16, KCNJ10 and PTH1R), medullary thick ascending limb (CLDN10) and ascending thin limb. Consistently, 3 groups of identified cells were also shown in the snATAC-seq dataset, suggesting that thick ascending limb subpopulations can be defined by either transcription or chromatin accessibility profiles. Furthermore, the authors detected a subset of proximal tubule cells with increased expression as chromatin accessibility of VCAM1 (2% of total cells) which also express Vimentin, CD24, and CD133, indicating that VCAM1 positive cell cluster may represent an injured or regenerative subpopulation.

Overall this is an important study, which could serve as reference for many studies to come. To qualify as reliable reference there needs to be some additional validation of the bioinformatics data though.

Specific comments:

Based on the ChromVAR motif activity analysis, authors have identified several genes, which are targeted either by HNF4A or TFAP2B. These are interesting and meaningful findings. However, a chromatin-immunoprecipitation qPCR experiment is required to validate identified interactions..

A Cis-coaccessibility network is reported, indicating that families of chromatin-chromatin interactions regulate gene expression by approximating enhancers and promoters. The authors should provide some verification for proposed interactions, i.e. apply epigenetic modulation approaches such as dCas9-DNMT1 or dCas9-KRAB to alter the chromatin accessibility of HNF4 promoter region to verify correlation of gene expression and chromatin accessibility.

There should be some additional validation of identified cell populations in the kidney.

Were the samples processed in the same time or in different bathes? Was batch correction analysis performed to minimize the batch effect?

The authors should provide a more detailed analysis information regarding the trajectory analysis with Cicero.

Reviewer #2 (Remarks to the Author):

In the manuscript entitled "Single cell transcriptional and chromatin accessibility profiling redefine cellular heterogeneity in the adult human kidney", Muto and colleagues collect paired single nucleus RNA-seq and ATAC-seq data on 5 human adult kidney samples. They use information from both data sets to define cell types and they take a deeper dive into cells of the thick ascending

limb and the proximal tubule. As my expertise is in single cell genomics and not kidney biology, my comments will be largely restricted to this domain. Overall, the data appear to be of sufficient quality and the analysis rests firmly in conventional methods. However, as one of only a few instances in the literature of paired data of this nature, I felt the authors could do more to directly integrate the two data types. In addition, while the sample size is small, there is an opportunity to explore genetic variation in this population that the authors did not address. Finally, some of the figures were not up to the standard, and some of the methods were insufficiently described. Below I address the major and minor criticisms:

Major criticisms

- 1) Integration of data. The authors do use the standard “integration” strategy from the Seurat pipeline to annotate cell types in addition to pulling a few anecdotes where change in transcription factor activity is correlated with change in TF expression level. However, this is one of only a handful of existing data sets where both types of data have been collected on the same samples and an opportunity to explore how complementary the data really are is missed here. One could use changes in expression across cell types and changes in co-accessibility to evaluate how well Cicero does in identifying promoter-enhancer links. One could look in a global way at how well TF expression level is correlated with TF activity level (from chromVAR). Are there factors that have low correlation? Or negative correlation? One could explore ways to overlay the pseudotime trajectories to get a better handle on direct and secondary targets of TFs in these differentiation pathways. There are a lot of directions left to explore the combination of the two data types that are left on the table here. It would be a shame not to push this a little further given the relative novelty of the data.
- 2) The authors make the case that the ATAC data may be a better cell type definer than the RNA data. To my knowledge this is different than what is generally observed in other data sets. It would be worthwhile to further explore this. In particular, the number of cells collected in the two data sets is significantly different (there are ~25% more cells in the ATAC data set). To support this observation, the authors should subset both data sets to the same number of cells (probably the same number of cells from each donor) and confirm that ATAC is still more informative. In addition, it would be interesting to look at how consistent the cell type proportions are across individuals for the two data types. Furthermore, the authors integrate with mouse data in one section and they acknowledge that mouse kidney atlases of RNA and ATAC exist. How comparable are the cell type compositions of mouse samples and human samples? Do differences tell us anything about biological differences between the two species?
- 3) The sample size here is obviously too small to do any association testing, but there are genetic differences between donors and the authors don't do anything to address whether genetic variation might be impacting gene regulation. However, methods exist to explore the impact of genetic variation on gene expression and chromatin in small samples (even in individual samples) – so-called allele-specific expression analysis and allele-specific hypersensitivity analysis. I would like to see the authors explore this further and at least comment on whether the relative influence of genetic variation seems consistent with published results in other tissues/cell lines.
- 4) QC Measures and Figures. Finally, there aren't sufficient QC metrics presented to really evaluate the quality of the data. How many cells were loaded in each 10X lane? How many reads were sequenced per cell? What was the estimated complexity? What was the fraction of reads in cells like? What did the distribution of fraction of reads in peaks look like? And what did the distribution of number of UMIs per cell and number of genes per cell look like? Etc. In addition, many of the main figures had inappropriately small labels, missing labels, etc. Specific critiques of figures will be listed in the minor criticisms below.

Minor criticisms

Lines 50-92: The logic of the Intro didn't flow that smoothly to me. I would recommend starting with the third paragraph (lines 68-77), then the first paragraph (lines 51-59). I would incorporate lines 78-82 into this new second paragraph. Then lines 60-67, and finally lines 83-91.

Lines 85 – 88: More important than the interactive website would be data matrices of read counts by gene or peak and metadata tables that include your final cell type assignments that people could download to explore the analyses on their own and to allow for label transfer to future projects.

Line 95/Fig. 1a: The schematic under “Multimodal single cell analysis” appears to just be a Cicero output. Please replace with something that indicates integrated analysis.

Line 101: The authors reference the R package used for analysis in the rest of the Results, so they should call out Seurat specifically here. Also, for Figure 1b, I'd like to see either a legend for the colors or have the cell type annotation text colored by the cluster it's defining.

Line 114: Here again, I would recommend the authors explicitly state that they used Seurat for label transfer.

Line 122: The axes labels for Fig. 1d are unreadable in the "pre-integration" UMAPs. Also, the same issue of legend or text color applies for the labels for cell type annotation in the last panel and Fig. 1e as in Fig. 1b.

Lines 122-124: I'd like to see boxplots of cell type proportions by individual for both technologies (RNA and ATAC) in the supplement. I'd also like to see the comparison after subsampling to the same number of cells.

Lines 131-134: As I mentioned in the major criticisms section, to make these claims about differences in power to observe cell types, the authors should subsample the data so that they have the same number of cells in the two analyses at a minimum.

Lines 149-152: The significance threshold for identifying differential expression or accessibility needs to be defined here and in the Methods. What FDR? What fold-change?

Lines 173-180: As mentioned in major criticisms, this section would be greatly improved by comparing globally the correlation structure between activity scores and gene expression.

Lines 182-183: The definition of CCANs is not accurate. CCANs just identify clusters of sites that co-vary. We infer that this implies a regulatory interaction. Please re-phrase.

Lines 192-198: The scale of Supp. Fig 7 are a little misleading to me. I would plot on a scale of 0 to 1, which will make the difference look much smaller. It's unclear to me how a paired t-test was implemented here. I think a Fisher's Exact Test or Mann-Whitney U test might be more appropriate.

Lines 202-204: This manuscript is not dedicated to evaluating the robustness of Cicero. I would just remove this last sentence of the paragraph.

Lines 207-217: Most of this is a discussion of the literature. The authors could more briefly introduce the topic here and move much of this section to the Intro or Discussion.

Line 227: Was the differential activity significant? Please state how that is defined.

Line 228: For Supp Fig. 9e, please change the color scheme so that it is not red and green (for people who are color blind).

Line 253: How many genes are differentially expressed at what FDR?

Lines 414 and 432: In these sections, we need to know how many cells were loaded on each lane. Also, what was the read configuration for each? How many cycles for Read 1 and Read 2?

Line 454: What was the FDR for significance? Was there a fold-change threshold as well?

Lines 467-468: What was the FDR for differential activity?

Line 487: What FDR for DARs?

Lines 495-497: FDR for differential expression? Same with Lines 500-502.

Line 730: I think you mean UMAP plot (singular). Same with Line 737.

Lines 734, 739: Define what's plotted in the dot plots more explicitly.

Lines 742-743: Be more explicit about what the color scale represents in Fig. 2a. Also label the color ramp in 2a. Is this a z-score?

Line 743: The axis label for the Y axis in the right panel is way too small.

Line 744: Pie charts are not appropriate ways to display data. I recommend converting 2b to a stacked bar chart and merging with 2c.

Line 748: Please label the color ramp in 3a and define in the figure legend. Same for 3b.

Line 750: Please define the Y axes for 3c. Make sure they are clearly labelled in the figure as well.

Line 754: I find the circus plots here and in Fig. S8 distracting. A barplot, or an UpSet plot, or a tanglegram would be more straightforward here.

Lines 758,765: Axes labels in a and c are way too small.

Reviewer #3 (Remarks to the Author):

Using multi-omics integrated analysis approach, the authors show unique cell states within the kidney and redefines cellular heterogeneity in the proximal tubule and thick ascending limb. The authors claim to have identified a novel renal progenitor cell type that are CD24+, CD133+ and VCAM+. They further claim that NF- κ B plays a role in the maintenance of these cells, which may be of clinical interest in designing therapies for acute kidney injury. In addition, through

multimodal approach, the authors demonstrate the heterogeneity within the thick ascending limb of loop of Henle at transcriptomic and chromatin accessibility level. Some of the key findings of this integrated analysis including the presence of a novel progenitor cells in adult human kidney is of great interest to nephrologist community. Moreover, this integrated study is first of its kind and was able to successfully identify and integrate complimentary features using computational tools. However, there are some concerns in this study, they are

1. Authors did provide immunofluorescence evidence for VCAM+ proximal cells. However, it would have been very helpful if authors had included CD24, CD133 along with VCAM1 in the immunofluorescence study. This is critical because the PT_VCAME cluster that the authors identified could in fact be a heterogeneous cell cluster. Based on the supplementary data, VCAM1 is expressed only in 39.6% of the cells in PT_VCAME cluster.
2. A key renal cell type that is missing in the cell clusters is the descending loop of Henle (DTL) cells. Studies have shown VCAM1 expression in this cell type. Due the proximity of these cell types to proximal cells, the authors have to make sure that novel cell cluster does not contain DTL cells.
3. In vitro study in which NFkB signaling was induced by TNF alpha in RPTEC does not directly provide evidence for the enrichment of this signaling pathway in Proximal_VCAME or for the transition of proximal cells to PT_VCAME state.
4. Supporting validations for some of the other key computational findings in the study would have significantly increased the impact of the study. For example, experimental validation of the regulatory role of HNF1B in CLDN10 expression.

New data for the revision:

Figures 4b, 4c, 4d, 5e and 5f

Supplementary Figures 5a, 5b, 5c, 5d, 5e, 5f, 8a, 8b, 8c, 8d, 8e, 9a, 9b, 10a, 10b, 10c, 10d, 10e, 10f, 14, 15c, 19a, 19b, 21a, 21b, 21c, 21d, 21e, 21f, 22a and 22b

Supplementary Tables 5, 6 and 7.

Supplementary Data 4, 7 and **Source Data** were also added.

Reviewer #1 (Remarks to the Author):

In this manuscript, Muto et al., utilized single nucleus RNA sequencing (snRNA-seq) and single nucleus ATAC sequencing (snATAC-seq) simultaneously to study the transcriptional and chromatin accessibility landscape of the adult human kidney. Joint profiling by scRNA-seq and snATAC-seq provided a framework for understanding how chromatin accessibility regulates transcription and this is the first study describing the single-cell epigenomic landscape of the human kidney. The authors applied this integration approach to highlight functional heterogeneity in the proximal tubule and thick ascending limb. They demonstrate that snATAC-seq provided additional information as compared to scRNA-seq and snATAC-seq.

The authors applied previously established R packages such as “chromVAR” to infer transcription-factor-associated chromatin accessibility, which means individual cell types are classified by transcription factor activity. The results are in agreement with previous finding that HNF4A binding motifs are enriched within differentially accessible chromatin regions (DAR) in the proximal tubule. Next, the authors applied the R package “Cicero” to predict cis-regulatory chromatin interactions for individual cell types. In order to study the cellular heterogeneity in the thick ascending limb, the authors performed unsupervised clustering analysis on the thick ascending limb in snRNA-seq dataset and identified 3 groups of cells: cortical thick ascending limb (CLDN16, KCNJ10 and PTH1R), medullary thick ascending limb (CLDN10) and ascending thin limb. Consistently, 3 groups of identified cells were also shown in the snATAC-seq dataset, suggesting that thick ascending limb subpopulations can be defined by either transcription or chromatin accessibility profiles. Furthermore, the authors detected a subset of proximal tubule cells with increased expression as chromatin accessibility of VCAM1 (2% of total cells) which also express Vimentin, CD24, and CD133, indicating that VCAM1 positive cell cluster may represent an injured or regenerative subpopulation.

Overall this is an important study, which could serve as reference for many studies to come. To qualify as reliable reference there needs to be some additional validation of the bioinformatics data though.

[Response] We thank the reviewer for the careful evaluation of our manuscript and for the positive comments. Our specific responses to the points raised follow:

- 1. Based on the ChromVAR motif activity analysis, authors have identified several genes, which are targeted either by HNF4A or TFAP2B. These are interesting and meaningful findings. However, a chromatin-immunoprecipitation qPCR experiment is required to validate identified interactions.*

[Response] We selected primary renal proximal tubule epithelial cells (RPTEC) to perform our validation. HNF4A expression was detectable in RPTEC, however, at a much lower level than kidney cortex (**new Supplementary Fig. 9b**). This limited our sensitivity to detect changes in chromatin accessibility for HNF4A.

New Supplementary Fig. 9b and 19a: Representative immunostaining images of HNF4A (red) in the kidney cortex or primary RPTEC, and VCAM1 (green) in the primary RPTEC. Scale bar indicates 100 μm .

We were able to validate HNF4A binding within open chromatin regions of multiple differentially expressed genes (SLC34A1, SLC5A2 and HNF4A) using ChIP followed by quantitative polymerase chain reaction (qPCR) analysis (ChIP-PCR) in RPTEC. We found that open chromatin regions with HNF4A motifs that were predicted to interact with a promoter via a cis-coaccessibility network were significantly enriched for HNF4A binding within the SLC34A1 and SLC5A2 locus when compared to a negative control (**new Supplementary Fig. 9a**). Again, the relatively low enrichment likely reflects the low absolute expression of HNF4A in cultured RPTEC.

- Promoter or first exon
- Open chromatin region co-accessible with the promoter
- Background control

Target	Coordinates
Background control	chr5:177393216-17739337
SLC34A1 promoter	chr5:177384377-177384389
SLC34A1 co-accessible	chr5:177379077-177379089
SLC5A2 promoter	chr16:31482945-31482957
SLC5A2 co-accessible	chr16:31458141-31458153
HNF4A first exon	chr20:44401447-44401459
HNF4A co-accessible	chr20:44395660-44395672

New Supplementary Fig. 9a: ChIP followed by quantitative PCR (ChIP-qPCR) analysis of HNF4A binding within the promoter, first exon or the open chromatin regions that were predicted to interact with a promoter via a CCAN in the differentially expressed gene loci (SLC34A1, SLC5A2 and HNF4A) in RPTEC (n = 3). ChIP-qPCR was performed with an open chromatin region without HNF4A motif on the intronic region of SLC34A1 gene as a background control. Data are mean±s.d. *P<0.05 (two-sided one sample t-test).

In addition to HNF4A, RPTEC expresses a pro-inflammatory marker (VCAM1) (**new Supplementary Fig. 19a**). We hypothesize that VCAM1 expression in tissue culture reflects an injury/dedifferentiation response of RPTEC grown on hard plastic. Although we do not expect RPTEC to fully capture the biology of PT_VCAM1, we used it as a model to investigate the role of RELA (NF- κ B) in predicted cis-regulatory interactions. We observed that an open chromatin region located 60 KB distal to the VCAM1 promoter that has a RELA motif predicted to interact with the promoter (via a cis-coaccessibility network) was enriched for RELA binding near the VCAM1 locus by ChIP-PCR (**new Fig.5f, Supplementary Fig. 19**).

New Supplementary Fig. 19b Graphical abstract of experimental methodology for RELA ChIP-qPCR. **New Fig. 5f**: ChIP followed by quantitative PCR (ChIP-qPCR) analysis of RELA binding within the promoter or the open chromatin region that was predicted to interact with a nearby promoter via a CCAN in the VCAM1 locus in RPTEC (n = 3). The background control was set on the region without RELA motif at the upstream of VCAM1 promoter. Data are means \pm s.d. *P<0.05 (two-sided one sample t-test).

Collectively, these findings are consistent with our hypothesis that HNF4A and RELA bind open chromatin regions and that chromVAR motif activity can predict genes targeted by specific transcription factors. Unfortunately, there is no suitable cell line for TAL or DCT, so we could not validate TFAP2B binding. However, there are existing reports that AP-2 family transcription factor (*Tfap2a* and *Tfap2b*) conditional knockout in mice affects distal convoluted tubule development [Chambers et al., PMID:31160420, Wang et al., PMID:29804851]. We have now addressed these points in the revised manuscript (**page 6 line 14-19, line 23-25; page 9 line 30-36**).

2. A *Cis-coaccessibility network* is reported, indicating that families of chromatin-chromatin interactions regulate gene expression by approximating enhancers and promoters. The authors should provide some verification for proposed interactions, i.e. apply epigenetic modulation approaches such as *dCas9-DNMT1* or *dCas9-KRAB* to alter the chromatin accessibility of *HNF4* promoter region to verify correlation of gene expression and chromatin accessibility.

[Response] As suggested by the reviewer, we attempted to apply *dCas9-KRAB* to alter accessibility of the *HNF4A* promoter and additional chromatin regions that were predicted to interact with the promoter via a *cis-coaccessibility network*. We performed lentiviral transduction of guide RNA cloned into CRISPR-repression optimized vector (CROP-seq-opti) and Lenti-*dCas9-KRAB*-blast in RPTEC, and confirmed transduction of *dCas9-KRAB* (**Fig. R1, for reviewing purposes only**).

Fig. R1: Representative immunostaining images of *dCas9-KRAB* fusion protein (red) in primary RPTEC with or without lentiviral transduction. The data were obtained 48 h after lentiviral transduction. Cells were fixed with 4% PFA, blocked with 1% bovine serum albumin, permeabilized with 0.1% Triton-X100 in PBS and incubated for 1 h with primary antibodies for CRISPR-Cas9 Antibody (clone 7A9-3A3, Novus Biologicals) followed by staining with secondary antibodies (Cy3-conjugated, Jackson ImmunoResearch). Scale bar indicates 100 μm .

We investigated the effects of CRISPR interference on HNF4A expression in RPTEC, but our results were indeterminate due to the low expression of HNF4A in cultured RPTEC. We hypothesize that dCas9-KRAB and guide RNA transduction may adversely affect HNF4A expression because RPTEC are sensitive to stress responses, including lentiviral transduction. In order to achieve 100% transduction we also had to add spinfection, which may have further caused RPTEC injury/dedifferentiation, reducing HNF4A expression even further.

We have made some progress towards developing this technique and have successfully decreased HNF4A expression in HepG2 cells (**Fig. R2**) using gRNA targeted to the promoter or first exon of HNF4A. Similarly, we were able to downregulate PTEN expression in either HepG2 or RPTEC (as a positive control). However, gRNA targeted to distal sites predicted to regulate HNF4A expression had no effect. These results are difficult to interpret because HepG2 cells likely do not have the same cis-regulatory architecture as RPTEC. We appreciate the reviewers' constructive suggestions and hope to further develop these techniques for future studies but are currently limited by the absence of human proximal tubule cell lines that retain their differentiated state in culture. We discussed the potential of epigenetic modulation approaches such as dCas9-DNMT1 or dCas9-KRAB to validate snATAC-seq data in the revised manuscript (**page 12 line 27- 29**).

Fig. R2: RT and real-time PCR analysis of mRNAs for HNF4A or PTEN in HepG2 cells or RPTEC infected with dCas-KRAB and guide RNA. The data were obtained from 50,000 cells 96 h after lentiviral transduction. Guide RNAs were designed for HNF4A promoter, 1st exon and open chromatin regions predicted to interact with the promoter in CCAN (distal 1-4) with CHOPCHOP (<https://chopchop.cbu.uib.no/>), and cloned into Crop-seq-opti. Guide RNAs for PTEN were used as positive controls (Lalli et al., PMID:32887689). NT, Nontargeting control gRNA.

3. *There should be some additional validation of identified cell populations in the kidney.*

[Response] We obtained additional lines of evidence to validate cell populations in the kidney. For PT_VCAM1, we performed immunostaining of human adult kidney cortex and found that VCAM1+ tubular cells were observed in 4.19 +/- 1.58% of LTL+ PT cells, whereas no VCAM1+ cells were detected in UMOD+ TAL cells in the kidney cortex (**new Fig. 4b**). These data suggest that PT_VCAM1 is a subpopulation of PT cells. Importantly, co-staining VCAM1 with AQP1 excluded the possibility that they might be mainly dTL cells (**new Fig. 4c**). Furthermore, costaining of CD24 or CD133 with VCAM1 on the human kidney sections revealed PT_VCAM1 is also a heterogeneous subset (**new Fig. 4d**). For validation of TAL subpopulations, we provide additional data showing that PTH1R and KCNJ10, which were predicted to be expressed in a subset of TAL, were detected in a subset of TAL (**New Supplementary Fig.14c**). We have now addressed these points in the revised manuscript (**page 7, line 43- page 8, line 24; page 8, line 32 - 38; page 8, line 44 - page 9, line 2**).

4. *Were the samples processed in the same time or in different bathes? Was batch correction analysis performed to minimize the batch effect?*

[Response] We processed samples at different time points (both nuclei prep and library construction). Batch correction was performed with the R package "Harmony" using the "RunHarmony" function in Seurat for both snATAC-seq and snRNA-seq datasets. We have now addressed these points in the revised manuscript (**page 4, line 6-7; page 13, line 40-41; page14, line 27-28**).

5. *The authors should provide a more detailed analysis information regarding the trajectory analysis with Cicero.*

[Response] We have expanded the description in the original manuscript. First, the CDS was constructed from the peak count matrix in the Seurat object of aggregated snATAC-seq data with "make_atac_cds" function with binarize = F. Next, the cds was preprocessed (num_dim = 50), aligned to remove batch effect (Laleh et al., PMID: 29608177) and reduced onto a lower dimensional space with the "reduce_dimension" function (reduction_method = 'UMAP', preprocess_method = "Aligned"). After filtering low quality cells, the cells were clustered (cluster_cells) and visualized. Subsequently, the dataset was subsetted for the PCT, PST and PT_VCAM1 clusters to perform cell ordering with the learn_graph function. We used the "order_cell" function and indicated three of the most distant cells from PT_VCAM1 in PCT and PST as "start points" of the trajectories. The data were visualized with "plot_accessibility_in_pseudotime" or "plot_cells" functions. Code is available in our online repository (https://github.com/p4rkerw/Muto_Wilson_NComm_2020). We have now addressed these points in the revised method section of manuscript (**page 15, line 26-36**).

Reviewer #2

In the manuscript entitled “Single cell transcriptional and chromatin accessibility profiling redefine cellular heterogeneity in the adult human kidney”, Muto and colleagues collect paired single nucleus RNA-seq and ATAC-seq data on 5 human adult kidney samples. They use information from both data sets to define cell types and they take a deeper dive into cells of the thick ascending limb and the proximal tubule. As my expertise is in single cell genomics and not kidney biology, my comments will be largely restricted to this domain. Overall, the data appear to be of sufficient quality and the analysis rests firmly in conventional methods. However, as one of only a few instances in the literature of paired data of this nature, I felt the authors could do more to directly integrate the two data types. In addition, while the sample size is small, there is an opportunity to explore genetic variation in this population that the authors did not address. Finally, some of the figures were not up to the standard, and some of the methods were insufficiently described. Below I address the major and minor criticisms:

[Response] We thank the reviewer for the careful evaluation of our manuscript and for the experimental suggestions that we feel have helped us to greatly improve our paper. Our specific responses to the points raised are as follows:

Major criticisms

1. Integration of data. The authors do use the standard “integration” strategy from the Seurat pipeline to annotate cell types in addition to pulling a few anecdotes where change in transcription factor activity is correlated with change in TF expression level. However, this is one of only a handful of existing data sets where both types of data have been collected on the same samples and an opportunity to explore how complementary the data really are is missed here. One could use changes in expression across cell types and changes in co-accessibility to evaluate how well Cicero does in identifying promoter-enhancer links. One could look in a global way at how well TF expression level is correlated with TF activity level (from chromVAR). Are there factors that have low correlation? Or negative correlation? One could explore ways to overlay the pseudotime trajectories to get a better handle on direct and secondary targets of TFs in these differentiation pathways. There are a lot of directions left to explore the combination of the two data types that are left on the table here. It would be a shame not to push this a little further given the relative novelty of the data.

[Response] We agree with reviewer 2 that data integration is a critical aspect of this manuscript that would benefit from further exploration and have incorporated multiple new analyses. In particular, we have explored gene-enhancer pairs, transcription factor expression and activity, pseudotemporal ordering of in the distal nephron and allele-specific expression. We have outlined our response to each of the major criticisms below:

1A. One could use changes in expression across cell types and changes in co-accessibility to evaluate how well Cicero does in identifying promoter-enhancer links.

[Response] To examine the relationship between Cicero co-accessibility and gene expression, we generated a plot of Cicero gene activity score relative to average log fold-change of cell-specific gene expression for all cell types (**Fig. R3 left: Cicero Gene Activity vs. Gene Expression**). Overall, we observed poor correlation between Cicero gene activity and gene

expression (Pearson $r^2=0.12$). These observations are consistent with previous reports that Cicero gene activity is a modest estimate of gene expression (Pliner et al PMID: 30078726). Interestingly, a minority of genes showed significant positive correlation between Cicero gene activity and gene expression (**Fig. R3 right: DACH1**), however, this was not typical. We have incorporated these data into our section on chromatin accessibility, TF activity, and chromatin interaction networks.

Fig. R3: Correlation between Cicero gene activity and gene expression across cell types.

Subsequently, we narrowed our analysis to predicted Cicero connections between enhancers and promoters. We annotated ATAC peaks to determine which peaks overlap with FANTOM enhancers (Andersson et al PMID:24670763) and then filtered Cicero connections for those that contain an enhancer-promoter pair. We compared enhancer-promoter co-accessibility to gene expression across cell types (**Fig. R4**). Similar to what we observed for Cicero gene activity scores, mean enhancer-promoter gene activity did not correlate well with gene expression (Pearson $r^2=0.075$). A limitation of this analysis is that available annotation databases do not have high-quality enhancer maps of the adult human kidney. Future analyses may benefit from kidney cell-type-specific profiling of enhancer-associated histone modifications (eg. H3K27ac).

Fig. R4: Correlation between enhancer-promoter co-accessibility and gene expression across all cell types.

1B. One could look in a global way at how well TF expression level is correlated with TF activity level (from chromVAR). Are there factors that have low correlation? Or negative correlation?

[Response] We generated a plot to examine the relationship between TF expression and TF chromVAR activity (**new Supplementary Fig. 8a**) and observed a modest correlation between the two variables (Pearson $r^2=0.36$).

New Supplementary Fig. 8a: Correlation between transcription factor chromVAR activity and gene expression across all cell types

However, as reviewer 2 suggested, it's likely that some transcription factors are acting as transcriptional repressors (rather than activators) and are negatively correlated with gene expression. Subsequently, we separated transcription factors into positively-correlated and negatively-correlated groups (**new Supplementary Fig. 8b,c**). Within each group, the correlation improved for both the positively-correlated and negatively-correlated TF relative to the overall comparison. Among the 452 TF in the JASPAR database, 11 TF showed significant negative correlation with gene expression and 38 showed significant positive correlation (Pearson $p<0.05$) across all kidney cell types. In contrast, some TF (**Fig. R5: HIF1A**) showed no correlation between expression and chromVAR activity.

New Supplementary Fig. 8b: Positively-correlated transcription factor motif – gene combinations across all cell types.

New Supplementary Fig. 8c: Negatively-correlated transcription factor motif – gene combinations across all cell types.

Fig. R5: Example of a transcription factor motif (HIF1A) that does not correlate with gene expression.

Interestingly, closely-related TF with important biologic function in the kidney may act in opposition to each other. For example, NR3C1 encodes the glucocorticoid receptor and NR3C2

encodes the mineralocorticoid receptor. The sequence motifs for these TF are extremely similar (Fornes et al. PMID:31701148, JASPAR2020), however, they have different roles. Aldosterone is the ligand for the mineralocorticoid receptor, which is abundantly expressed in the distal nephron and serves to regulate sodium reabsorption. Importantly, the mineralocorticoid receptor can also be activated by corticosteroids, which are the ligand for glucocorticoid receptors. Specific enzymes are expressed in the distal nephron (eg. HSD11B2) that metabolize corticosteroids to prevent activation of the mineralocorticoid receptor. In this manner, NR3C1 and NR3C2 can be thought of as having opposing effects. In our data set, we observed a positive correlation between NR3C1 expression and chromVAR motif activity, whereas NR3C2 showed a negative correlation (**new Supplementary Fig. 8d,e**).

New Supplementary Fig. 8d,e: (d) Correlation between glucocorticoid receptor (NR3C1) motif activity and gene expression. (e) Correlation between mineralocorticoid receptor (NR3C2) motif activity and gene expression

These data may reflect cell-specific chromatin accessibility profiles that bias the cellular response to closely-related TF. We have now addressed these points in the revised manuscript (**page 5, line 41 -page 6, line 10**).

1C. One could explore ways to overlay the pseudotime trajectories to get a better handle on direct and secondary targets of TFs in these differentiation pathways

[Response] We employed pseudotemporal ordering within the distal nephron to identify TF or signaling pathways associated with different cell fates. Some of the cell types in the distal nephron include distal convoluted tubule (DCT), connecting tubule (CNT), and principal cells (PC), which are closely-related cell types oriented in series. Pseudotemporal ordering of the snATAC data for these cell types identified chromatin regions that either open or close from proximal to distal (**new Supplementary Fig. 10a,b**).

New Supplementary Fig. 10a,b: Pseudotemporal ordering of the distal nephron snATAC dataset. PC-principal cells, CNT-connecting tubule, DCT – distal convoluted tubule

We performed a transcription factor motif enrichment analysis of pseudotime-dependent peaks to identify enriched transcription factors. Among the 452 TF motifs that we evaluated, we identified 24 TF that were significantly enriched (FDR < 0.05) with a fold-enrichment greater than two. Within this family of enriched TF, we identified TFAP2B, which we previously-described as associated with distal nephron fate. In addition, we identified a candidate list of TF (ZBTB33, CREB3, E2F1, etc.) that potentially regulate distal nephron fate. Subsequently, we performed pseudotemporal ordering of the distal nephron cells in our snRNA dataset (**new Supplementary Fig. 10c,d**) to determine if these TF are differentially expressed or if there is enrichment of downstream signaling pathways.

New Supplementary Fig. 10c,d: Pseudotemporal ordering of the distal nephron snRNA dataset. PC-principal cells, CNT-connecting tubule, DCT – distal convoluted tubule

We clustered the pseudotime-dependent genes into gene modules to determine which modules of genes are preferentially expressed as we progress from proximal to distal nephron (**new**

Supplementary Fig. 10e). We plotted three gene modules (6,8,12) to visualize where they lie along the pseudotemporal trajectory. Module 6 has the highest activity in PC and CNT among all the gene modules, and it was selected for further interrogation (**new Supplementary Fig. 10f**)

New Supplementary Fig. 10e,f: Pseudotime-dependent gene modules that are significantly up- or down-regulated in the distal nephron progressing from proximal to distal.

We intersected the genes in module 6 with our list of enriched transcription factors obtained from pseudotemporal ordering of the snATAC data to determine if any of the enriched TF also showed differential expression. As a result, we obtained a short list of TF that change both their expression and motif accessibility (EGR1, TFAP2A, TFAP2C, ZSCAN4, STAT1, STAT3, KLF9, NR2F2, IRF1) during the transition from distal convoluted tubule to principal cell within the distal nephron. Not surprisingly, several of the top GO enriched pathways for the genes within module 6 include 1) regulation of metanephric nephron tubule epithelial cell differentiation 2) positive regulation of hormone metabolic process and 3) collecting duct development. We have now addressed these points in the revised manuscript (**page 6, line 25-41**).

2. The authors make the case that the ATAC data may be a better cell type definer than the RNA data. To my knowledge this is different than what is generally observed in other data sets. It would be worthwhile to further explore this. In particular, the number of cells collected in the two data sets is significantly different (there are ~25% more cells in the ATAC data set). To support this observation, the authors should subset both data sets to the same number of cells (probably the same number of cells from each donor) and confirm that ATAC is still more informative. In addition, it would be interesting to look at how consistent the cell type proportions are across individuals for the two data types. Furthermore, the authors integrate with mouse data in one section and they acknowledge that mouse kidney atlases of RNA and ATAC exist. How comparable are the cell type compositions of mouse samples and human samples? Do differences tell us anything about biological differences between the two species?

[Response] We have outlined our response to each of the major criticisms below:

2A. To support this observation, the authors should subset both data sets to the same number of cells (probably the same number of cells from each donor) and confirm that ATAC is still more informative

[Response] We have subsampled our data as Reviewer 2 suggested to determine if the snATAC dataset is able to distinguish between the proximal convoluted tubule (PCT) and proximal straight tubule (PST) when it has the same number of cells as the snRNA dataset. The initial distribution of cells in the snRNA and snATAC datasets by donor is depicted in Table R1 and the distribution after subsampling is shown in Table R2.

Table 1		
snRNA	snATAC	donor
4114	4166	Control_1
2920	4072	Control_2
5464	7381	Control_3
3683	5280	Control_4
3804	6135	Control_5
19985	27034	

Table 2 – Downsample		
snRNA	snATAC	donor
4114	4114	Control_1
2920	2920	Control_2
5464	5464	Control_3
3683	3683	Control_4
3804	3804	Control_5
19985	19985	

The downsampled snATAC dataset was able to clearly distinguish between PCT and PST (**new Supplementary Fig. 5a**).

New Supplementary Fig. 5a: Differentiating between the proximal convoluted tubule (PCT) and proximal straight tubule (PST) in a downsampled snATAC dataset.

To further explore the ability of snATAC to resolve the difference between PCT and PST we further subsampled the snATAC dataset to 50% total cells compared to the snRNA dataset as detailed in Table R3.

Table R3 – Downsample 50%		
snRNA	snATAC	donor
4114	2057	Control_1
2920	1460	Control_2
5464	2732	Control_3
3683	1841	Control_4
3804	1902	Control_5
19985	9992	

New Supplementary Fig. 5b: Differentiating between the proximal convoluted tubule (PCT) and proximal straight tubule (PST) in a downsampled snATAC dataset with 50% as many cells as the snRNA dataset.

The snATAC subset with half as many cells is still able to resolve the difference between PCT and PST (**new Supplementary Fig. 5b**). We have now addressed these points in the revised manuscript (**page 4, line 43- page 5, line 2**).

2B. In addition, it would be interesting to look at how consistent the cell type proportions are across individuals for the two data types.

[Response] The below are the tables for the distribution of our snRNA and snATAC datasets by cell type and donor. The box-and-whisker plots were included in the **new Supplementary Fig. 22** and the tables were in the **Source data**.

Distribution of snATAC dataset by cell type and donor						
snATAC	Control_1	Control_2	Control_3	Control_4	Control_5	Total
PCT	629	1352	2015	1163	1109	6268
PST	355	429	1558	977	961	4280
PT_VCAM1	83	137	247	83	124	674

PEC	137	71	41	71	83	403
TAL	1203	450	1858	1967	2284	7762
DCT	511	825	638	368	435	2777
CNT	260	169	152	96	221	898
PC	324	177	251	222	328	1302
ICA	106	121	181	44	159	611
ICB	227	106	124	46	117	620
PODO	14	23	30	12	56	135
ENDO	170	58	181	180	170	759
MES_FIB	81	100	69	34	68	352
LEUK	66	54	36	17	20	193
Total	4166	4072	7381	5280	6135	27034

Distribution of snRNA dataset by cell type and donor						
snRNA	Control_1	Control_2	Control_3	Control_4	Control_5	Total
PT_VCAM1	59	66	213	70	41	449
PT	765	1190	1433	613	1035	5036
PEC	178	54	121	65	134	552
TAL	1017	393	628	1995	402	4435
DCT1	721	365	714	257	704	2761
DCT2	97	63	75	65	189	489
CNT	336	272	631	124	442	1805
PC	131	199	368	109	215	1022
ICA	153	124	546	75	209	1107
ICB	150	30	93	19	57	349
PODO	93	34	182	72	82	463
ENDO	312	100	290	156	150	1008
MES	70	19	67	36	47	239
FIB	22	3	84	22	76	207
LEUK	10	8	19	5	21	63
Total	4114	2920	5464	3683	3804	19985

2C. Furthermore, the authors integrate with mouse data in one section and they acknowledge that mouse kidney atlases of RNA and ATAC exist. How comparable are the cell type compositions of mouse samples and human samples? Do differences tell us anything about biological differences between the two species?

[Response] In our experience with single cell profiling in mouse and human kidney tissue, the proportion of each cell type is heavily influenced by sample type (fresh vs frozen), dissociation method (single cell vs. single nucleus), and library preparation technique (drop-seq vs. 10X). It is difficult to directly compare cell type proportions between human and mouse samples because human samples are heterogeneous and subject to sampling bias. For example, the relative quantity of cortex vs. medulla can have a huge effect on the observed proportion of cells. Previously-published mouse scRNA atlases (Ransick et al PMID: 31689386, Park et al. PMID:

29622724) do not include a table of relative cell type proportions or cell barcode annotations, but appear roughly equivalent to our human and mouse datasets. That being said, it may be more instructive to compare healthy human kidney to diseased human kidney (or healthy to diseased mouse) to determine if the proportion of cells change or if new cell types emerge.

3. The sample size here is obviously too small to do any association testing, but there are genetic differences between donors and the authors don't do anything to address whether genetic variation might be impacting gene regulation. However, methods exist to explore the impact of genetic variation on gene expression and chromatin in small samples (even in individual samples) – so-called allele-specific expression analysis and allele-specific hypersensitivity analysis. I would like to see the authors explore this further and at least comment on whether the relative influence of genetic variation seems consistent with published results in other tissues/cell lines.

[Response] We agree with Reviewer #2 that genetic variation plays a critical role in the regulation of gene expression and have examined allele-specific expression in our dataset. We followed GATK best practices (Poplin et al <https://doi.org/10.1101/201178>) to identify heterozygous germline variants within coding regions using HaplotypeCaller. We implemented the WASP allele specific pipeline (Geijn et al PMID: 26366987) to mitigate read mapping bias prior to generating allele specific read counts with GATK ASEReadCounter. Subsequently, we examined allele-specific expression using ASEP (Fan et al PMID: 32392242), which is a method that performs gene-based allele-specific expression analysis across a population. In our first experiment, we incorporated all cell types into a 'pseudo-bulk' sample and examined reads that mapped to coding transcripts (5'UTR, exons, 3'UTR). In the pseudo-bulk dataset we had 401 genes that met our filtering criteria to examine allele-specific expression. Among these 401 genes, we identified 84 that showed allele-specific expression after Benjamini-Hochberg adjustment for multiple comparisons (FDR<0.05). A subset of these genes have important functions in the kidney, including *CLCNKB* and *SLC12A3* (new **Supplementary Fig. 14**).

New Supplementary Fig. 14: Allele-specific expression of *CLCNKB* and *SLC12A3* among all cell types in the snRNA dataset.

We compared our list of pseudo-bulk allele-specific genes to a previously-published list of allele-specific genes obtained from bulk RNA-seq of human kidney (Fan et al PMID: 32392242). Fan et al reported a total of 304 genes that showed allele-specific expression among the 6,540

genes they analyzed. Ten of the allele-specific genes in our dataset were also reported in their dataset. Subsequently, we demultiplexed our dataset using our cell type barcode annotations to investigate allele-specific expression in individual cell types. Within the proximal tubule, there were 77 genes that met filtering criteria and 17 showed allele-specific expression. The majority of the proximal tubule allele-specific genes (n=12/17) were also detected in our pseudo-bulk dataset.

In our second experiment, we expanded the analysis to include reads that map to introns. Our samples were obtained from nuclear preps and a significant proportion of our reads map to unspliced pre-mrna. This approach increased the number of genes that met our filtering criteria to 1430 in the pseudobulk analysis. Among these genes, we observed 432 with allele-specific expression. Overall, our estimate of the proportion of genes that show allele-specific variation ranged from 20.9 to 30.2%. This estimate is significantly greater than the 4.6% reported by Fan et al using bulk RNA-seq. Notably, Fan et al reported that sequencing depth has a significant effect on the sensitivity of their method to detect allele-specific expression. Their dataset contained 121 tubule compartment samples sequenced to a median depth of 35 million reads. In contrast, our samples were sequenced to a mean depth of 377 million reads. In addition, our samples were obtained from a nuclear preparation and captured the entire kidney cortex rather than just the tubulointerstitium. These factors may partially explain the difference in our ability to detect allele-specific expression. Genes that were evaluated for allele-specific expression were now listed as **new Supplementary Data 4**. We have now addressed these points in the revised manuscript (**page 7, line 19- 42; page 16, line 8 -32**).

4. *QC Measures and Figures. Finally, there aren't sufficient QC metrics presented to really evaluate the quality of the data. How many cells were loaded in each 10X lane? How many reads were sequenced per cell? What was the estimated complexity? What was the fraction of reads in cells like? What did the distribution of fraction of reads in peaks look like? And what did the distribution of number of UMIs per cell and number of genes per cell look like? Etc. In addition, many of the main figures had inappropriately small labels, missing labels, etc. Specific critiques of figures will be listed in the minor criticisms below.*

[Response] We have included all of the requested QC metrics in our manuscript (method section) or supplementary figures. Please see a detailed response below.

4A. *How many cells were loaded in each 10X lane?*

[Response] 10,000 cells were loaded in each 10X lane (**page 13, line 19; page 14, line 12**).

4B. *How many reads were sequenced per cell?*

[Response] A mean of 377,573,305 reads (SD = 76,365,483) were sequenced for each snRNA library corresponding to a mean of 70,886 reads per cell (SD=8633). A mean of 318,097,692 reads were sequenced for each snATAC library (SD=54,357,210) corresponding to a mean of 12,946 fragments per cell (SD=2,960). See table below. (**page 14, line 16-18, new Supplementary table 5**).

Donor	Total RNA reads	RNA reads per cell	Total ATAC reads	ATAC fragments per cell
Healthy_1	499801345	72383	343687555	13892
Healthy_2	317710661	74844	269154397	12611
Healthy_3	391611498	59344	396525222	17493
Healthy_4	368420549	81944	314024252	10567
Healthy_5	310322473	65915	267097032	10168
mean	377573305.2	70886	318097691.6	12946.2
stdev	76365483	8632	54357209	2960

4C. *What was the estimated complexity? What was the fraction of reads in cells like?*

[Response] The library complexity for the snRNA libraries was estimated with sequencing saturation. The mean sequencing saturation was 81.4 +/- 2.4 %. The mean fraction of reads with a valid barcode (fraction of reads in cells) was 88.2 +/- 5.9%. See Table below. (**page 13, line 35-36, new Supplementary table 6**).

Quality Control for snRNA Libraries		
Donor	Sequencing Saturation	Fraction reads with Valid Barcode
Healthy_1	77.5	89.6
Healthy_2	83.4	77.9
Healthy_3	83	92.5
Healthy_4	82.6	91.7
Healthy_5	80.5	89.7
mean	81.4	88.28
stdev	2.4	5.9

The mean sequencing saturation for snATAC libraries was 37.3 +/- 2.2 % and the mean fraction of reads with a valid barcode was 97.3 +/- 1.2 %. See table below. (page 14, line 18-19, new Supplementary table 7).

Quality control for snATAC libraries		
Donor	Sequencing Saturation	Fraction reads with Valid Barcode
Healthy_1	36.2	98.3
Healthy_2	35.1	98.3
Healthy_3	37.1	98.3
Healthy_4	41.1	95.8
Healthy_5	37.3	95.8
mean	37.36	97.3
stdev	2.2	1.3

4D. *What did the distribution of fraction of reads in peaks look like?*

[Response] Fraction of reads in peaks, number of reads in peaks per cell and ratio reads in genomic blacklist region per cell were included (new Supplementary Fig. 21). We have now addressed these points in the revised manuscript (page 14, line 24- 26).

New Supplementary Figure 21 – QC metrics for snATAC-seq dataset: (d) Fraction of reads in peaks, (e) number of reads in peaks per cell and (f) ratio of reads in genomic blacklist region per cell in snATAC-seq data were shown.

4E. And hat did the distribution of number of UMIs and genes per cell look like?

[Response] Number of genes per cell, number of UMIs per cell and fraction of mitochondrial genes per cell were included (**new Supplementary Fig. 21**). We have now addressed these points in the revised manuscript (**page 13, line 42- page14 line 2**).

New Supplementary Figure 21 – QC metrics for snRNA- dataset: (a) Number of genes per cell, (b) number of UMIs per cell and (c) fraction of mitochondrial genes per cell in snRNA-seq data were shown.

Minor criticisms

Lines 50-92: The logic of the Intro didn't flow that smoothly to me. I would recommend starting with the third paragraph (lines 68-77), then the first paragraph (lines 51-59). I would incorporate lines 78-82 into this new second paragraph. Then lines 60-67, and finally lines 83-91.

[Response] We have made the suggested changes to the introduction.

Lines 85 – 88: More important than the interactive website would be data matrices of read counts by gene or peak and metadata tables that include your final cell type assignments that people could download to explore the analyses on their own and to allow for label transfer to future projects.

[Response] Read count matrices have been uploaded to GSE151302 and include outputs from cellranger (snRNA-seq) and cellranger-atac (snATAC-seq) data. For snATAC data, the filtered peak barcode matrix and fragments files are made available. For snRNA data, the filtered feature barcode matrix file is made available. Additional cellranger outputs (eg. Coordinate-sorted bam files) and metadata are made available through the Human Cell Atlas www.humancellatlas.org.

Line 95/Fig. 1a: The schematic under “Multimodal single cell analysis” appears to just be a Cicero output. Please replace with something that indicates integrated analysis.

[Response] – We have made the requested change to Figure 1A.

Line 101: The authors reference the R package used for analysis in the rest of the Results, so they should call out Seurat specifically here. Also, for Figure 1b, I'd like to see either a legend for the colors or have the cell type annotation text colored by the cluster it's defining.

[Response] We have made the requested edits to specifically mention Seurat. We have modified Figure 1B to change the annotation text colors.

Line 114: Here again, I would recommend the authors explicitly state that they used Seurat for label transfer.

[Response] – We have explicitly mentioned Seurat as suggested.

Line 122: The axes labels for Fig. 1d are unreadable in the “pre-integration” UMAPs. Also, the same issue of legend or text color applies for the labels for cell type annotation in the last panel and Fig. 1e as in Fig. 1b.

[Response] – We have modified the figures as suggested.

Lines 122-124: I'd like to see boxplots of cell type proportions by individual for both technologies (RNA and ATAC) in the supplement. I'd also like to see the comparison after subsampling to the same number of cells.

[Response] A table of cell type proportions by individual for snRNA and snATAC libraries is included in response 2B along with boxplots. The comparison after subsampling to the same number of cells is shown in response 2A. These data are included as Supplementary Tables 2 and 3 along with the boxplots.

Lines 131-134: As I mentioned in the major criticisms section, to make these claims about differences in power to observe cell types, the authors should subsample the data so that they have the same number of cells in the two analyses at a minimum.

[Response] The comparison after subsampling to the same number of cells is shown in response 2A.

Lines 149-152: The significance threshold for identifying differential expression or accessibility needs to be defined here and in the Methods. What FDR? What fold-change?

[Response] The significance threshold for both differential expression and differential accessibility is defined as a Bonferroni adjusted p-value < 0.05 . Similarly, the positive (and negative) fold-change threshold is the natural log of 0.25 with an additional requirement that at least 20% of cells express the target gene (or ATAC peak). These definitions have been added to the main text and methods sections.

Lines 173-180: As mentioned in major criticisms, this section would be greatly improved by comparing globally the correlation structure between activity scores and gene expression.

[Response] See response to major criticism 1.

Lines 182-183: The definition of CCANs is not accurate. CCANs just identify clusters of sites that co-vary. We infer that this implies a regulatory interaction. Please re-phrase.

[Response] We have rephrased the definition of CCANs as suggested.

Lines 192-198: The scale of Supp. Fig 7 are a little misleading to me. I would plot on a scale of 0 to 1, which will make the difference look much smaller. It's unclear to me how a paired t-test was implemented here. I think a Fisher's Exact Test or Mann-Whitney U test might be more appropriate.

[Response] We have regenerated the figure with an updated scale as suggested (**Supplementary Fig. 11**). To compare the proportion of Cicero connections identified in the GeneHancer database, we created a contingency table by Cicero coaccess threshold to include the number of connections identified in the GeneHancer database (In GH) and the number not in the database (Not in GH). We performed a chi-squared test and the result was highly significant for an association between coaccess threshold and membership in the GeneHancer database (p val < 2.2e-16). The table was included in the **Source Data**.

Cicero Connections in All Cell Types Present in GeneHancer Double Elite Stratified by Cicero Coaccess Threshold		
Threshold	In GH	Not in GH
0.1	642906	303176
0.2	311546	138076
0.3	166256	70038
0.4	89334	36922
0.5	47360	18784
0.6	23972	9168
0.7	10714	3822
0.8	3484	1108
Pearson's chi-squared=1520.8, p-value < 2.2e-16		

Lines 202-204: This manuscript is not dedicated to evaluating the robustness of Cicero. I would just remove this last sentence of the paragraph.

[Response] We have removed this sentence as suggested.

Lines 207-217: Most of this is a discussion of the literature. The authors could more briefly introduce the topic here and move much of this section to the Intro or Discussion.

[Response] We have moved the majority of these lines to the discussion as suggested.

Line 227: Was the differential activity significant? Please state how that is defined.

[Response] – Differential activity for HNF1B and ESRRB was assessed with the FindMarkers function using the chromVAR assay with default parameters by comparing each TAL subcluster to the remaining TAL cells. Bonferroni-adjusted p-values were used to assess FDR ($p_{adj} < 0.05$, **Supplementary Fig. 15f**). Subsequently, we used the Seurat FindMarkers function to identify differentially accessible chromatin regions that differentiate between TAL1 and TAL2. We performed a transcription factor motif enrichment of these DAR using the Seurat FindMotifs function and ranked the top-enriched motifs in open chromatin regions for both cell types (**Supplementary Fig. 15g**).

Line 228: For Supp Fig. 9e, please change the color scheme so that it is not red and green (for people who are color blind).

[Response] We have changed the color scheme as suggested.

Line 253: How many genes are differentially expressed at what FDR?

[Response] A total of 463 genes were differentially expressed between PT and PT_VCAM1 at an FDR < 0.05 and log-fold-change threshold of +/- 0.25.

Lines 414 and 432: In these sections, we need to know how many cells were loaded on each lane. Also, what was the read configuration for each? How many cycles for Read 1 and Read 2?

[Response] A target of 10,000 nuclei were loaded onto each lane for both snRNA and snATAC libraries. For snRNA, libraries 1-3 were sequenced with a 2x100bp and libraries 4-5 were sequenced with 2x150bp paired-end configuration. The cDNA for snRNA libraries was amplified for 17 cycles. For snATAC, libraries 1-3 were sequenced with a 2x50bp and libraries 4-5 were sequenced with a 2x150bp paired-end configuration. Sample index PCR was performed at 12 cycles. We have now addressed these points in the revised manuscript (**page 13, line 31-32; page 14 line 14-15**).

Line 454: What was the FDR for significance? Was there a fold-change threshold as well?

[Response] The FDR for significance for all comparisons is $\text{padj} < 0.05$. The fold-change threshold is 0.25.

Lines 467-468: What was the FDR for differential activity?

[Response] The FDR is Bonferroni $\text{padj} < 0.05$.

Line 487: What FDR for DARs?

[Response] The FDR is Bonferroni $\text{padj} < 0.05$.

Lines 495-497: FDR for differential expression? Same with Lines 500-502.

[Response] The FDR is Bonferroni $\text{padj} < 0.05$.

Line 730: I think you mean UMAP plot (singular). Same with Line 737.

[Response] Changes made.

Lines 734, 739: Define what's plotted in the dot plots more explicitly.

[Response] We have made changes to the figure legend accordingly.

Lines 742-743: Be more explicit about what the color scale represents in Fig. 2a. Also label the color ramp in 2a. Is this a z-score?

[Response] The color scale represents a z-score of the number of Tn5 sites within each DAR. We have labeled the color ramp in Fig.2a.

Line 743: The axis label for the Y axis in the right panel is way too small.

[Response] Axis labels have been enlarged

Line 744: Pie charts are not appropriate ways to display data. I recommend converting 2b to a stacked bar chart and merging with 2c.

[Response] We agree that the pie chart does not add much additional information and have eliminated it from the figure.

Line 748: Please label the color ramp in 3a and define in the figure legend. Same for 3b.

[Response] We have updated the figure and defined the color scale in the legend.

Line 750: Please define the Y axes for 3c. Make sure they are clearly labelled in the figure as well.

[Response] We have defined the Y axes for 3c as peak coverage and Cicero co-accessibility score.

Line 754: I find the circus plots here and in Fig. S8 distracting. A barplot, or an UpSet plot, or a tanglegram would be more straightforward here.

[Response] We prefer to leave the circos plot in Fig. 3d (and Supplementary Fig. 8) unchanged because we feel that it best demonstrates links between predicted chromatin interactions. For comparison, we have included a representative barplot for predicted interactions in the PCT (Fig. R6) and we feel that this is less intuitive than a circos plot. Similarly, we do not think that a tanglegram or UpSet plot would better convey the data.

Lines 758,765: Axes labels in a and c are way too small.

[Response] We have enlarged the axes labels as suggested.

Reviewer #3

Using multi-omics integrated analysis approach, the authors show unique cell states within the kidney and redefines cellular heterogeneity in the proximal tubule and thick ascending limb. The authors claim to have identified a novel renal progenitor cell type that are CD24+, CD133+ and VCAM+. They further claim that NF- κ B plays a role in the maintenance of these cells, which may be of clinical interest in designing therapies for acute kidney injury. In addition, through multimodal approach, the authors demonstrate the heterogeneity within the thick ascending limb of loop of Henle at transcriptomic and chromatin accessibility level. Some of the key findings of this integrated analysis including the presence of a novel progenitor cells in adult human kidney is of great interest to nephrologist community. Moreover, this integrated study is first of its kind and was able to successfully identify and integrate complimentary features using computational tools. However, there are some concerns in this study, they are:

[Response] We appreciate these positive comments and thank the reviewer. Our point by point response to the specific concerns follows:

1. Authors did provide immunofluorescence evidence for VCAM+ proximal cells. However, it would have been very helpful if authors had included CD24, CD133 along with VCAM1 in the immunofluorescence study. This is critical because the PT_VCAM cluster that the authors identified could in fact be a heterogeneous cell cluster. Based on the supplementary data, VCAM1 is expressed only in 39.6% of the cells in PT_VCAM1 cluster

[Response] As suggested by the reviewer, we co-stained VCAM1 with CD24 or CD133 in acetone-fixed sections of adult human kidney cortex samples. We found that a subset of VCAM1+ proximal tubular cells expresses CD24 or CD133 (**new Fig. 4d**). We also observed that CD24+ or CD133+ cells are rare in VCAM1- proximal tubular cells. As Reviewer #3 suggested, these findings are consistent with the idea that the PT_VCAM1 cluster is a heterogeneous population. We have now addressed these points in the revised manuscript (**page 8, line 44 - page 9, line 2**).

New Fig. 4d: Representative immunostaining images of CD24 or CD133 (red) and VCAM1 (green) in the cortex of adult human kidney. Arrowheads indicate VCAM1 co-expression with CD24 or CD133 in PT and arrows mark VCAM1 expression without CD24 or CD133. Scale bar indicates 20 μm .

2. A key renal cell type that is missing in the cell clusters is the descending loop of Henle (DTL) cells. Studies have shown VCAM1 expression in this cell type. Due the proximity of these cell types to proximal cells, the authors have to make sure that novel cell cluster does not contain DTL cells.

[Response] We stained kidney sections with AQP1 to localize the PT, DTL and descending vasa recta. We observed that VCAM1 was expressed in a small subset of the DTL (**new Fig. 4c**).

New Fig. 4c: Representative immunohistochemical images of SLC34A1, UMOD or AQP1 (blue) and VCAM1 (brown) expression in the adult human kidneys. An arrowhead marks VCAM1 expression in the DTL and an arrow marks DTL without VCAM1 expression. Scattered brown dots are seen with multiple different antibodies and considered non-specific staining. Scale bar indicates 50 μm .

We also confirmed that VCAM1+ tubular cells were observed in 4.19 +/- 1.58% of LTL+ PT cells, whereas no VCAM1+ cells were detected in UMOD+ TAL cells in the kidney cortex (**new Fig. 4b**). These data suggest that the majority of VCAM1+ tubular cells are in the PT within the cortex. A small minority of DTL tubules expressed VCAM1 compared to a larger number VCAM1+ PT cells. Based on these findings, we concluded that VCAM1+ DTL cells are unlikely to be a significant portion of the cells in our dataset. We have now addressed these points in the

New Fig. 4b: Immunofluorescence staining for VCAM1 (green), UMOD (red) and LTL (white) in the adult human kidney sections (left, representative image) and quantitation of VCAM1-positive cells on the LTL-positive cells or UMOD-positive cells (right). The sections were quantified in 5 random fields for each patient (n = 3). Arrowheads indicate VCAM1-positive cells in the LTL-positive PT. Scale bar indicates 100 μm . Box-and-whisker plots depict the median, quartiles and range. ***P<0.001 (Student's t test).

revised manuscript (page 8, line 32 - 38).

3. *In vitro* study in which NF κ B signaling was induced by TNF alpha in RPTEC does not directly provide evidence for the enrichment of this signaling pathway in Proximal_VCAM1 or for the transition of proximal cells to PT_VCAM1 state.

[Response] As the reviewer points out, *in vitro* studies do not provide direct evidence for the enrichment of this signaling pathway in PT_VCAM1 or for the transition of PT to PT_VCAM1. We have removed this figure and related statement. Although the NF- κ B signaling pathway was significantly enriched in PT_VCAM1 in both the snRNA-seq and snATAC-seq analysis (**Figure 4e and Supplementary Fig. 18**), we could not detect RELA, RELB, or their phosphorylated forms in PT_VCAM1 cells (**Table. R7**). We hypothesize that NF- κ B expression may be too low to detect with our methods. As a result, we cannot exclude the potential role of NF- κ B signaling in the transition of PT to PT_VCAM1.

A second transcription factor motif whose activity was predicted to be significantly changed in PT_VCAM1 compared to normal PT was HNF4A. We observed that HNF4A protein expression was lost in PT_VCAM1 nuclei (**new Fig. 5g**). These data are consistent with the predicted decrease in HNF4A motif activity. We have now addressed these points in the revised manuscript (**page 9, line 25- 30**).

Table R7: REAGENT	SOURCE	IDENTIFIER
NF- κ B p65 Rabbit mAb (D14E12)	Cell Signaling Technology	8242T
NF- κ B p65 Mouse mAb (L8F6)	Cell Signaling Technology	6956T
NF- κ B p65 Mouse mAb	Millipore	17-10060
p-NF- κ B p65 Antibody mouse mAb (27.Ser 536)	Santa Cruz Biotechnology	sc-136548
RELB Mouse mAb (17.3)	Invitrogen	437500
RELB Rabbit mAb (D7D7W)	Cell Signaling Technology	10544
p-RELB (Ser552) Rabbit mAb (D41B9)	Cell Signaling Technology	5025S

New Fig. 5g: Immunofluorescence staining for VCAM1 (green), HNF4A (red) and LTL (white) in the adult human kidney sections (left, representative image) and quantitation of HNF4A-positive cells on the VCAM1-positive or negative subset of LTL-positive PT cells (right). The quantitation is shown in Figure 5g.

VCAM1 / HNF4A / LTL

4. Supporting validations for some of the other key computational findings in the study would have significantly increased the impact of the study. For example, experimental validation of the regulatory role of *HNF1B* in *CLDN10* expression.

[Response] In response to this comment, we provide new data obtained from human kidney samples to validate heterogeneity within the TAL. PTH1R and KCNJ10 were both predicted to be expressed in a subset of TAL cells. We used immunohistochemistry to demonstrate that PTH1R and KCNJ10 were expressed in a subset of UMOD+ SLC12A1+ cells (**new Supplementary Fig. 14c**).

KCNJ10 / UMOD

PTH1R / UMOD

KCNJ10 / SLC12A1

PTH1R / SLC12A1

New Supplementary Fig. 14c: Representative immunohistochemical images of KCNJ10 or PTH1R (brown) and UMOD or SLC12A1 (blue) in the adult human kidneys. Scale bar indicates 50 μm .

We initially described these subsets as medullary vs cortical TAL based on CLDN10 and CLDN16 mRNA expression. However, our new data suggests that both of these TAL subsets exist in the cortical TAL, and we have changed our terminology to "TAL1" and "TAL2". We have addressed these points in the revised manuscript (**page 7, line 43- page 8, line 24**).

In general, few TAL cell lines exist. Some rodent medullary TAL cell lines have been established previously (Eng et al., PMID: 17670898, Bourgeois et al., PMID: 12836026), although they were unlikely to keep all the characteristics of mature TAL cells due to loss of a complex physiological organization in vivo. Actually, murine Hnf1b regulates Cldn10b, Cldn19, and Cldn3 differently in one of these cell models in vitro than it does in vivo in mouse kidney (Kompatscher et al., PMID: 29561186). Hence, we were not able to experimentally validate the regulatory role of HNF1B in CLDN10 expression for now. We are sorry not to be able to respond to this constructive suggestion of the reviewer, but we will take this advice into consideration for our future studies. As a result, we have removed the related statement about the potential relationship between HNF1B and CLDN10 (**page 10, line 44- page 11, line 5**).

Instead, to validate some of our other key computational findings, we additionally performed ChIP-qPCR to validate HNF4A and RELA motif analysis (**new Fig. 5f, Supplementary Fig.9**). We have now addressed these points in the revised manuscript (**(page 6 line 14-19, line 23-25; page 9 line 30-36)**).

REVIEWERS' COMMENTS

Reviewer #1 (Remarks to the Author):

The authors took ask my concerns to heart and provide extensive new data and sufficient explanations. I have no further comments.

Reviewer #3 (Remarks to the Author):

The authors have tried to address the comments either by validating or changing the text in the manuscript. Although the authors were not able to address all of the reviewers 'comments, the study does include two key findings that may have clinical implications. These include:

1. The finding of PT_VCAM1 as a subpopulation of proximal tubular cells that is undergoing injury in situ, and expands in aging and chronic kidney disease.
- 2.Role of NF- κ B in the maintenance of PT_VCAM1.

Reviewer #4 (Remarks to the Author):

In this manuscript, Yoshiharu Muto et al profiled multiple human kidneys with 10x based single nucleus RNA and ATAC-seq. By integrating these two datasets, this study revealed unique biological insight including cell type-specific gene expression and regulation mechanisms by cis- and trans- factors. Also, this research characterized cell state heterogeneity in the thick ascending limb, as well as a unique proximal tubule cell subtype that is related to tubular injury, which is novel and could be foundations for therapy development to treat kidney injury.

I am satisfied with the high data quality, detailed analysis, and validation experiments. Also, the authors have satisfactorily responded to all major and minor comments from reviewer #2. A minor comment is that several supplementary analysis in S8 and S15 are pretty interesting and can be considered to be part of the main figure. Overall, I am pretty excited about the release of the dataset and believe the publication of this study will benefit both the single-cell genomic field and renal research.